# ACCELERATING NEURAL ODEs: A VARIATIONAL FORMULATION-BASED APPROACH

**Hongjue Zhao**[1], **Yuchen Wang**[2], **Hairong Qi**[3], **Zijie Huang**[4], **Han Zhao**[1], **Lui Sha**[1], **Huajie Shao**[2]
[1]University of Illinois Urbana-Champaign, [2]William & Mary, [3]University of Tennessee, Knoxville, [4]University of California Los Angeles
{hongjue2, hanzhao, lrs}@illinois.edu, {ywang142, hshao}@wm.edu
hqi@utk.edu, zijiehuang@cs.ucla.edu

## ABSTRACT

Neural Ordinary Differential Equations (Neural ODEs or NODEs) excel at modeling continuous dynamical systems from observational data, especially when the data is irregularly sampled. However, existing training methods predominantly rely on numerical ODE solvers, which are time-consuming and prone to accumulating numerical errors over time due to autoregression. In this work, we propose VF-NODE, a novel approach based on the variational formulation (VF) to accelerate the training of NODEs. Unlike existing training methods, the proposed VF-NODEs implement a series of global integrals, thus evaluating Deep Neural Network (DNN)–based vector fields only at specific observed data points. This strategy drastically reduces the number of function evaluations (NFEs). Moreover, our method eliminates the use of autoregression, thereby reducing error accumulations for modeling dynamical systems. Nevertheless, the VF loss introduces oscillatory terms into the integrals when using the Fourier basis. We incorporate Filon's method to address this issue. To further enhance the performance for noisy and incomplete data, we employ the natural cubic spline regression to estimate a closed-form approximation. We provide a fundamental analysis of how our approach minimizes computational costs. Extensive experiments demonstrate that our approach accelerates NODE training by 10 to 1000 times compared to existing NODE-based methods, while achieving higher or comparable accuracy in dynamical systems. The code is available at https://github.com/ZhaoHongjue/VF-NODE-ICLR2025.

## 1 INTRODUCTION

Neural ordinary differential equations (NODEs) (Chen et al., 2018) represent a family of continuous-depth machine learning models. Drawing inspiration from ResNets (He et al., 2016), NODEs aim to parameterize vector fields of ODEs using deep neural networks (DNNs),

$$\dot{\boldsymbol{x}} = \boldsymbol{f_\theta}(t, \boldsymbol{x}), \tag{1}$$

where $\boldsymbol{f_\theta} : [0, T] \times \mathbb{R}^d \to \mathbb{R}^d$ is a DNN and $\boldsymbol{\theta}$ denotes the model parameters. The continuous nature and specific inductive bias of NODEs render them particularly well-suited for modeling dynamical systems from *irregularly sampled time series* data (Rubanova et al., 2019; Kidger et al., 2020). Thus, NODEs have been widely applied to various dynamical system applications, such as multi-agent trajectory forecasting (Wen et al., 2022), model-based reinforcement learning (Alvarez et al., 2020), optimal control (Chi, 2024) and chemical reaction process modeling (Yin et al., 2023).

In existing training frameworks of NODEs, numerical ODE solvers play a crucial role. The forward pass outcomes are directly calculated using numerical ODE solvers. For the backward pass, there are two methods commonly employed to backpropagate through ODE solvers (Kidger, 2022; Onken & Ruthotto, 2020): (1) *discretize-then-optimize*, which involves directly backpropagating through operations of ODE solvers, and (2) *optimize-then-discretize*, also known as the *adjoint sensitivity method*, as utilized in (Chen et al., 2018), which introduces additional adjoint ODEs. In this method, gradients of the scalar loss function with respect to parameters of NODEs are computed by solving these adjoint ODEs. More details can be found in Appendix B.1.

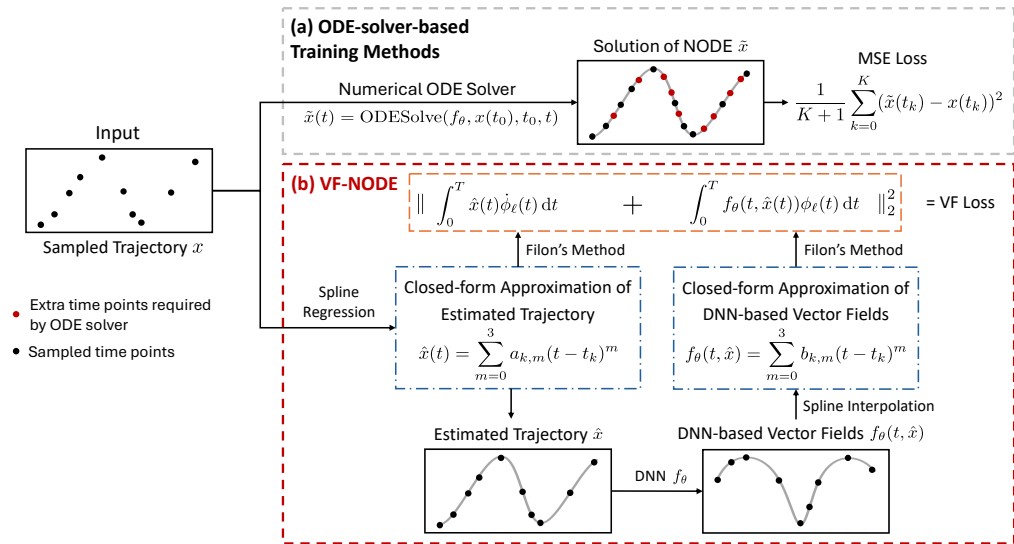

Figure 1: Comparison of VF-NODEs and ODE-solver-based methods. (a) ODE-solver-based training methods: ODE solvers compute solutions of the NODEs, necessitating evaluations of the DNN-based vector fields at *additional* data points. (b) VF-NODEs: implement global integrals numerically using Filon's method and spline regression in the VF loss, evaluating the DNN-based vector fields only at specific data points.

Nonetheless, these ODE-solver-based training methods face two significant limitations. First, they are inherently *time-consuming*. The internal mechanisms within numerical ODE solvers can incur significant computational costs in solving NODEs (Lipman et al., 2022). This is attributed to the numerous evaluations of DNN-based vector fields beyond given sampled data points, as shown in Fig. 1. For the optimize-then-discretize approach, this issue may be exacerbated by the introduction of additional adjoint ODEs. Second, existing approaches may suffer from low accuracy. On the one hand, the autoregressive nature of most numerical ODE solvers can lead to error accumulation, as discussed in Appendix B.2. On the other hand, the optimize-then-discretize approach *incurs additional numerical discretization error*, resulting in inaccurate gradients, and potentially causing the training process to fail entirely (Gholami et al., 2019). In short, these limitations stem from the use of *numerical ODE solvers*.

To date, various approaches have been proposed to address the aforementioned limitations of NODEs. For example, to alleviate the computational bottleneck of NODEs, some works attempt to constrain the complexity of learned dynamics (Finlay et al., 2020; Kelly et al., 2020; Pal et al., 2021), while others seek to directly modify the ODE-solver-based training process (Kidger et al., 2021; Djeumou et al., 2022; Norcliffe & Deisenroth, 2023; Matei et al., 2023). In the domain of irregularly sampled time series tasks, several models built upon neural differential equations have been proposed (Rubanova et al., 2019; Kidger et al., 2020). Nevertheless, these approaches remain heavily reliant on ODE solvers, preventing them from effectively addressing the computational bottleneck.

To address the challenges associated with the reliance on numerical ODE solvers during the training of NODEs, we propose the VF-NODE, a novel approach that employs the variational formulation (VF) (Brunel et al., 2014; Hackbusch, 2017) to accelerate training. The proposed approach integrates a VF-based loss function within the standard NODE architecture. Our *motivation* is that the VF loss can implement a series of *global integrals*, thus evaluating the DNN-based vector field only at specific data points in the observations. In contrast, existing ODE-solver-based training methods have to evaluate the DNN-based vector fields at extra data points due to step size settings. Consequently, our method significantly reduces the number of function evaluations (NFEs). On the other hand, these global integrals mitigate autoregression compared with ODE-solver-based training methods, eliminating error accumulation and improving prediction accuracy. However, the utilization of the VF loss poses two challenges for NODEs. (i) It could result in additional oscillatory integrals due to the introduction of Fourier basis functions like sine and cosine. (ii) It does not perform well when data is missing and noisy. To address the first challenge, we incorporate Filon's method (Deaño

et al., 2017) for oscillatory integrals. For the second challenge, we adopt the natural cubic spline regression (De Boor, 1978) to obtain a closed-form approximation for Filon's method. We also provide a fundamental analysis of the acceleration benefits of our approach in Section 4.3. Extensive experiments demonstrate that the VF-NODE achieves a 10 to 1000 times speed increase over existing baselines while maintaining higher or comparable accuracy.

**Our contributions are three-fold**: (1) We introduce a novel approach that employs the variational formulation (VF) to significantly accelerate the training of NODEs. This method significantly reduces the number of function evaluations (NFEs) and improves the prediction accuracy. (2) We integrate Filon's method with natural cubic spline regression to effectively compute oscillatory integrals from noisy and partially observed data within the VF loss. (3) Evaluation results on multiple dynamical systems including one real-world application demonstrate that our approach achieves 10 to 1000 times acceleration in training speed compared to the baselines while achieving higher or competitive accuracy.

## 2 RELATED WORK

**NODEs for Dynamical Systems.** NODEs have been introduced to model continuous dynamical systems. Latent ODEs based on VAEs were first proposed in (Chen et al., 2018), with RNNs as encoders and NODEs as decoders. Rubanova et al. (2019) proposed ODE-RNNs, which use NODEs to simulate continuous dynamics of hidden states in RNNs, and incorporated them into Latent ODEs as encoders. Kidger et al. (2020) introduced Neural Controlled Differential Equations (NCDEs), which can be viewed as the deep limit of RNNs. However, complex mechanisms involved in these methods exacerbate the computational burden during training. In contrast, our approach is built on common NODEs, requiring only simple multi-layer perceptrons (MLPs) and the VF loss. It can significantly speed up NODE training and improve the robustness against noisy data. Biloš et al. (2021) introduced Neural Flows as an alternative to NODEs, which directly use DNNs to model the solution of ODEs. However, they have difficulty modeling autonomous systems.

**Acceleration Techniques of NODEs.** Existing works on accelerating the NODEs training can be divided into three categories. (i) *Regularization-based methods.* Some approaches try to constrain the complexity of learned dynamics using regularization techniques, including high-order derivatives regularization (Kelly et al., 2020), kinetic regularization (Finlay et al., 2020), temporal regularization (Ghosh et al., 2020) and others. Unfortunately, these methods may only accelerate inference, and the training time may not be reduced. (ii) *Design new architectures.* Some works directly restrict the dynamics by designing special architectures of NODEs, for example, Heavy Ball NODEs (Xia et al., 2021) and Nesterov NODEs (Nguyen et al., 2022). In addition, model order reduction techniques have been utilized to directly compress DNNs in NODEs (Lehtimäki et al., 2022). However, these methods limit the expressivity of learned dynamics, making them unsuitable for dynamical systems. (iii) *Modify training process.* Other approaches try to modify the training process of NODEs to implement acceleration, such as the IRDM (Daulbaev et al., 2020), seminorm approach (Kidger et al., 2021), the Taylor theorem (Djeumou et al., 2022), the Gauß–Legendre quadrature (Norcliffe & Deisenroth, 2023), and the sensitivity-free gradient descent (Matei et al., 2023). Nevertheless, all existing modifications still rely on ODE solvers. Thus, the existing computational bottleneck is not tackled effectively, leading to poor acceleration performance. In contrast, our method only performs global numerical integrals, which is much more efficient than existing approaches.

**Variation Formulation of ODEs**. The VF method, as employed in our work, was initially introduced for parameter estimation of *known* ODEs (Brunel et al., 2014). Subsequent research, such as D-CODE (Qian et al., 2022), extended the use of VF for *symbolic regression* to facilitate the discovery of equations. This approach was primarily adopted to circumvent the need for numerically estimating derivatives from noisy data. However, it was not utilized to expedite the training of models, unlike in our work. Furthermore, the evaluation of D-CODE was limited to dynamical systems sampled at regular intervals, employing the basic compound trapezoidal rule (Press, 2007) for computing numerical integrals in the loss function. Notably, D-CODE did not address the challenges posed by oscillatory integrals, which arise with the Fourier basis function (Deaño et al., 2017). *In contrast, our application of VF to NODEs serves distinct purposes.* Firstly, we leverage VF to accelerate NODEs training. Secondly, we utilize it to avoid autoregression, thereby enhancing prediction accuracy. From an application perspective, our focus is on modeling dynamical systems based on

irregularly sampled data. On the technical front, we integrate Filon's method to effectively handle oscillatory integrals, a significant advancement over the D-CODE approach.

## 3 PRELIMINARIES

**Problem Setup.** Consider a time-series dataset in which each trajectory is represented as $\{(t_0, \boldsymbol{x}(t_0)), (t_1, \boldsymbol{x}(t_1)), \ldots, (t_K, \boldsymbol{x}(t_K))\}$, where $0 = t_0 < t_1 < \cdots < t_K = T$, and $\boldsymbol{x}(t_k) = [x_1(t_k), x_2(t_k), \ldots, x_d(t_k)]^\top \in (\mathbb{R} \cup \{*\})^d$ are noisy observations, with $*$ denoting possible missing values. Our goal is to speed up the training of NODEs for dynamical systems.

### 3.1 VARIATIONAL FORMULATION OF ODES

In this subsection, we formally introduce the variational formulation (VF) of ODEs we utilized in Theorem 1. Through this formulation, we can establish a direct connection between the trajectory $\boldsymbol{x}$ and the vector field $\boldsymbol{f}$ through a numerical integral.

**Theorem 1** (Variational Formulation of ODEs (Brunel et al., 2014; Hackbusch, 2017))**.** Consider $d \in \mathbb{N}^+$, $T \in \mathbb{R}^+$, continuous functions $\boldsymbol{x} : [0, T] \to \mathbb{R}^d$, $\boldsymbol{f} : [0, T] \times \mathbb{R}^d \to \mathbb{R}^d$, and $\phi \in \mathcal{C}^1[0, T]$, where $\mathcal{C}^1$ is the set of continuously differentiable functions. Here we define the functionals

$$\boldsymbol{c}(\boldsymbol{x}, \boldsymbol{f}, \phi) := \int_0^T \boldsymbol{x}(t)\dot{\phi}(t) \, \mathrm{d}t + \int_0^T \boldsymbol{f}(t, \boldsymbol{x}(t))\phi(t) \, \mathrm{d}t \,. \tag{2}$$

Then $\boldsymbol{x}$ is the solution to the ODEs $\dot{\boldsymbol{x}} = \boldsymbol{f}(t, \boldsymbol{x})$ if and only if

$$\boldsymbol{c}(\boldsymbol{x}, \boldsymbol{f}, \phi) = \boldsymbol{0}, \quad \forall \phi \in \mathcal{C}^1[0, T] \quad \text{s.t. } \phi(0) = \phi(T) = 0. \tag{3}$$

The proof of Theorem 1 can be found in (Qian et al., 2022). This theorem establishes the necessary and sufficient conditions under which $\boldsymbol{x}$ is the solution of the ODEs $\dot{\boldsymbol{x}} = \boldsymbol{f}(t, \boldsymbol{x})$. Moreover, in contrast to existing training frameworks for NODEs, this connection does not rely on solving the ODE numerically.

Intuitively, for a NODE expressed as Eq. (1), we can attempt to find the optimal parameters $\boldsymbol{\theta}$ by minimizing $\|\boldsymbol{c}(\boldsymbol{x}, \boldsymbol{f}, \phi)\|_2^2 = \sum_{j=1}^d c_j^2(\boldsymbol{x}, \boldsymbol{f}, \phi)$ to zero to satisfy Eq. (3). To operationalize this concept in NODE training, we introduce Theorem 2 (Qian et al., 2022), which is outlined below.

**Theorem 2.** Let $\boldsymbol{x} : [0, T] \to \mathbb{R}^d$ be a continuously differentiable function which satisfies $\dot{\boldsymbol{x}} = \boldsymbol{f}(t, \boldsymbol{x})$. Then for the Lipschitz continuous neural network $\boldsymbol{f}_{\boldsymbol{\theta}} : [0, T] \times \mathbb{R}^d \to \mathbb{R}^d$, where $\boldsymbol{\theta}$ are parameters, the following limit holds.

$$\lim_{L \to \infty} \sum_{\ell=1}^L c_j^2(\boldsymbol{x}, \boldsymbol{f}_{\boldsymbol{\theta}}, \phi_\ell) = \|(f_{\boldsymbol{\theta}, j} - f_j) \circ (t, \boldsymbol{x})\|_2^2, \tag{4}$$

where $\{\phi_1, \phi_2, \ldots, \phi_L\}$ are a series of Hilbert orthonormal basis for $L^2[0, T]$ such that $\phi_\ell(0) = \phi_\ell(T) = 0$, $\ell = 1, \ldots, L$ and $\|(f_{\boldsymbol{\theta}, j} - f_j) \circ (t, \boldsymbol{x})\|_2^2 = \int_0^T [f_{\boldsymbol{\theta}, j}(t, \boldsymbol{x}(t)) - f_j(t, \boldsymbol{x}(t))]^2 \, \mathrm{d}t$.

The proof of Theorem 2 can be found in Appendix C. In general, Theorem 2 enables the application of the VF to the training of NODEs. By leveraging this theorem, we convert an infinite number of constraints into a series of constraints using a set of orthogonal basis functions. Additionally, we establish a connection between the VF and distances between functions, implying that as $\|\boldsymbol{c}(\boldsymbol{x}, \boldsymbol{f}_{\boldsymbol{\theta}}, \phi)\|_2^2$ converges to zero, the parameters $\boldsymbol{\theta}$ in NODEs also converge to their optimal values.

### 3.2 FILON'S METHOD

Filon's method is designed for oscillatory integrals. Consider the integral $\int_0^T h(t) \sin(\omega t) \, \mathrm{d}t$. We aim to compute this integral numerically using the available data points $\{h(t_k)\}_{k=0}^K$ with $0 = t_0 < t_1 < \cdots < t_K = T$.

For general numerical integration techniques, such as the Newton-Cotes formula (Press, 2007), they utilize polynomials $p_k(t)$ to approximate the entire integrand $h(t)\sin(\omega t)$ within each time interval $[t_k, t_{k+1}]$. Consequently, these methods struggle to handle high-frequency oscillations as $\omega$ increases, because polynomials cannot effectively match the high-order derivatives of the integrand. Further analysis on this issue is provided in Appendix D. To address this limitation, we introduce Filon's method (Deaño et al., 2017), which employs polynomials solely to approximate the non-oscillatory component $h(t)$ in the original integrand. This approach enables the estimation of the original integral as follows:

$$\int_0^T h(t)\sin(\omega t)\,\mathrm{d}t = \sum_{k=0}^{K-1} \int_{t_k}^{t_{k+1}} h(t)\sin(\omega t)\,\mathrm{d}t \approx \sum_{k=0}^{K-1} \int_{t_k}^{t_{k+1}} q_k(t)\sin(\omega t)\,\mathrm{d}t\,, \quad (5)$$

where $q_k(t)$ is the approximate $n$-th-order polynomial of $h(t)$ over the time interval $[t_k, t_{k+1}]$. It is important to note that the integral $\int_{t_k}^{t_{k+1}} q_k(t)\sin(\omega t)\,\mathrm{d}t$ can always be computed analytically. This approach ensures that the accuracy of numerical integrals is not affected by $\omega$, effectively addressing the challenge of oscillatory integrals.

## 4    PROPOSED METHOD

In this section, we begin by introducing the VF loss for NODEs based on Theorem 2 and addressing the computational challenges in the VF loss using Filon's method and natural cubic spline regression. Then, we elaborate on the steps of VF-NODEs. Finally, we provide a fundamental analysis of the acceleration benefits of our approach.

### 4.1    COMPUTING THE VF LOSS

In Section 3.1, we systematically introduced the VF utilized in this work, and demonstrated the connection between this formulation and the distance between the vector field $\boldsymbol{f_\theta}$ of the NODE and the ground truth vector field $\boldsymbol{f}$. Based on Theorem 1 and Theorem 2, the optimal parameters $\boldsymbol{\theta}$ of a NODE, as expressed in Eq. (1), can be obtained by solving the following optimization problem:

$$\boldsymbol{\theta}^\star = \arg\min_{\boldsymbol{\theta}} \sum_{i=1}^N \sum_{\ell=1}^L \left\| \boldsymbol{c}(\boldsymbol{x}^{[i]}, \boldsymbol{f_\theta}, \phi_\ell) \right\|_2^2,$$

$$\boldsymbol{c}(\boldsymbol{x}^{[i]}, \boldsymbol{f_\theta}, \phi_\ell) = \int_0^T \boldsymbol{x}^{[i]}(t)\dot{\phi}_\ell(t)\,\mathrm{d}t + \int_0^T \boldsymbol{f_\theta}(t, \boldsymbol{x}^{[i]}(t))\phi_\ell(t)\,\mathrm{d}t\,, \quad (6)$$

where $\{\boldsymbol{x}^{[i]}\}_{i=1}^N$ are trajectories in the dataset, and $\phi_\ell(t) = \sqrt{2/T}\sin(\pi\ell t/T)$, which can be considered as the sine Fourier basis. However, computing the VF loss poses two computational challenges. (i) The use of the sine Fourier basis introduces oscillatory terms into the integrals. (ii) It is challenging to compute integrals numerically from noisy and incomplete data.

To address the first challenge, we introduce the Filon's method, which is designed for oscillatory integrals. Integrals in Eq. (6) are one-dimensional oscillatory integrals, and these integrals can be concluded as $\int_0^T h(t)\sin(\omega t)\,\mathrm{d}t$, without loss of generality. As discussed in Section 3.2, polynomials struggle to capture the nature of the oscillatory integrand $h(t)\sin(\omega t)$ as $\omega$ increases. To tackle this issue, we leverage Filon's method, which uses polynomials to approximate the non-oscillatory part $h(t)$ in each interval. By focusing on $h(t)$ alone, we eliminate the influence of $\omega$ on the precision of the closed-form approximation.

To deal with the second challenge, we introduce the natural cubic spline regression (De Boor, 1978), which is used to build up a precise closed-form polynomial approximation for $h(t)$ in Filon's method from noisy data. For $\{h(t_k)\}_{k=0}^K$ with $0 = t_0 < t_1 < \cdots < t_K = T$, we aim to construct a cubic polynomial $q_k(t)$ in each interval $[t_k, t_{k+1}]$ in Eq. (5):

$$q_k(t) = a_{k,0} + \sum_{m=1}^3 a_{k,m}(t - t_k)^m = a_{k,0} + a_{k,1}(t - t_k) + a_{k,2}(t - t_k)^2 + a_{k,3}(t - t_k)^3. \quad (7)$$

By meeting the requirements for continuity, differentiability, natural boundary conditions, and smoothness, we can calculate the coefficients $a_{k,m}$ ($k = 0, \ldots, K-1, m = 0, 1, 2, 3$) in Eq. (7). In

general, spline regression allows us to create sufficiently precise *closed-form* approximations while maintaining smoothness. We provide a detailed process for this calculation in Appendix E.

In summary, to compute the integral $\int_0^T h(t)\sin(\omega t)\,\mathrm{d}t$, we first construct the natural cubic spline approximation of $h(t)$, denoted as $q_k(t)$, within each time interval using sampled data $\{(t_k, h(t_k))\}_{k=0}^K$. By substituting Eq. (7) into Eq. (5), we can estimate the integral accurately by computing $\sum_{k=0}^{K-1} \int_{t_k}^{t_{k+1}} q_k(t)\sin(\omega t)\,\mathrm{d}t$.

## 4.2 VF-NODEs

The proposed VF-NODE uses the same neural architecture as the vanilla NODE, but it is trained based on the VF loss. Given a trajectory $\{(t_k, \boldsymbol{x}(t_k))\}_{k=0}^K$, where $0 = t_0 < t_1 < \cdots < t_K = T$, we present the detailed steps of our method.

**Step 1**: Perform natural cubic spline *regression* on $\boldsymbol{x}(t_k)$ to get the spline coefficients $\boldsymbol{a}_{k,m}$.

**Step 2**: Estimate the trajectory based on the spline to remove noise: $\hat{\boldsymbol{x}}(t_k) = \boldsymbol{a}_{k,0} + \sum_{m=1}^3 \boldsymbol{a}_{k,m}(t_k - t_k)^m = \boldsymbol{a}_{k,0}$ $(k = 0, \ldots, K-1)$ and $\hat{\boldsymbol{x}}(t_K) = \sum_{m=0}^3 \boldsymbol{a}_{K-1,m}(t_K - t_{K-1})^m$.

**Step 3**: Evaluate the vector fields $\boldsymbol{f}_{\boldsymbol{\theta}}(t_k, \hat{\boldsymbol{x}}(t_k))$, $k = 0, \ldots, K$.

**Step 4**: Perform natural cubic spline *interpolation* on $\boldsymbol{f}_{\boldsymbol{\theta}}(t_k, \hat{\boldsymbol{x}}(t_k))$ to get the spline coefficients $\boldsymbol{b}_{k,m}$.

**Step 5**: Compute the VF loss $\|\boldsymbol{c}(\hat{\boldsymbol{x}}, \boldsymbol{f}_{\boldsymbol{\theta}}, \phi_\ell)\|_2^2$ based on $\boldsymbol{a}_{k,m}$ and $\boldsymbol{b}_{k,m}$:

$$\sum_{\ell=1}^L \|\boldsymbol{c}(\hat{\boldsymbol{x}}, \boldsymbol{f}_{\boldsymbol{\theta}}, \phi_\ell)\|_2^2 = \sum_{\ell=1}^L \left\| \int_0^T \hat{\boldsymbol{x}}(t)\dot{\phi}_\ell(t)\,\mathrm{d}t + \int_0^T \boldsymbol{f}_{\boldsymbol{\theta}}(t, \hat{\boldsymbol{x}}(t))\phi_\ell(t)\,\mathrm{d}t \right\|_2^2 \tag{8}$$

where

$$
\begin{aligned}
\int_0^T \hat{\boldsymbol{x}}(t)\dot{\phi}_\ell(t)\,\mathrm{d}t &= \sqrt{\frac{2}{T}}\frac{\pi\ell}{T} \sum_{k=0}^{K-1}\sum_{m=0}^3 \boldsymbol{a}_{k,m} \int_{t_k}^{t_{k+1}} (t - t_k)^m \cos\frac{\pi\ell t}{T}\,\mathrm{d}t\,, \\
\int_0^T \boldsymbol{f}_{\boldsymbol{\theta}}(t, \hat{\boldsymbol{x}}(t))\phi_\ell(t)\,\mathrm{d}t &\approx \sqrt{\frac{2}{T}} \sum_{k=0}^{K-1}\sum_{m=0}^3 \boldsymbol{b}_{k,m} \int_{t_k}^{t_{k+1}} (t - t_k)^m \sin\frac{\pi\ell t}{T}\,\mathrm{d}t\,.
\end{aligned}
\tag{9}
$$

These steps are summarized as Algorithm 1 in Appendix F. In this process, the natural cubic spline serves two key roles in our method. (i) It builds up precise closed-form approximations of the trajectory $\boldsymbol{x}(t)$, denoted as $\hat{\boldsymbol{x}}(t)$, and vector fields $\boldsymbol{f}_{\boldsymbol{\theta}}(t, \hat{\boldsymbol{x}}(t))$ from noisy observations, which is essential for estimating the oscillatory integral accurately. (ii) It fills in missing values in the trajectory $\boldsymbol{x}(t)$, which is necessary for evaluating the vector fields $\boldsymbol{f}_{\boldsymbol{\theta}}(t, \hat{\boldsymbol{x}}(t))$.

*Remark* 1. It is important to emphasize that our primary use of natural cubic spline is for constructing the closed-form approximation, which is essential for computing numerical integrals, rather than for transforming the original irregularly sampled data into regularly spaced data. While we do utilize this technique to fill in missing values, it is a natural byproduct of constructing the closed-form approximation.

*Remark* 2. Because we have removed noise in observations by performing spline *regression* on $\boldsymbol{x}(t)$, we only need to perform spline *interpolation* on $\boldsymbol{f}_{\boldsymbol{\theta}}(t, \hat{\boldsymbol{x}}(t))$ for obtaining a closed-form approximation.

## 4.3 Fundamental Analysis for the Acceleration of VF-NODEs

Now we demonstrate how our method accelerates the training of NODEs in two ways: (1) reduce the number of function evaluations (NFEs) and (2) improve parallelizability. Consider a trajectory with $M$ observed data points. As shown in Fig. 1 (a), ODE-solver-based training methods require evaluating vector fields $\boldsymbol{f}_{\boldsymbol{\theta}}$ many more than $M$ times. For example, common adaptive-step-size explicit Runge-Kutta methods (Butcher, 2016) use an autoregressive formula $\boldsymbol{x}_{n+1} = \boldsymbol{x}_n + h\sum_{j=1}^J z_j \boldsymbol{g}_j$ to estimate the solution of a NODE expressed as in Eq. (1). Here, $h$ is the step size, $J$ is the order

of the ODE solver, $z_j$ are coefficients and $g_j$ are evaluations of the DNN-based vector field $f_\theta$. In this case, to make one-step-forward prediction, the DNN-based vector field must be be evaluated for $J$ times. Moreover, to ensure accuracy, these ODE solvers often require evaluating the DNN-based vector field at additional data points beyond those provided in the observations. As a result, the vector field $f_\theta$ will be evaluated $\gg M \times J$ times using ODE-solver-based training methods. Additionally, due to the autoregressive nature of ODE solvers, these vector fields must be evaluated step by step.

However, VF-NODEs only evaluate the DNN-based vector fields for exactly $M$ times, as shown in Fig. 1 (b). As discussed in Section 4.2, to compute global integrals in the VF loss numerically, we evaluate the vector fields $f_\theta$ only at the specific sampled data points in **Step 3**, in order to construct a closed-form spline approximation in **Step 4**. In addition, these vector fields can be evaluated simultaneously in **Step 3**. Although $L$ integrals need to be computed in $c(x, f_\theta, \phi_\ell)$ for $\ell = 1, \ldots, L$, these integrals share the same vector fields, allowing for efficient computation.

**Summary of the Proposed VF-NODEs.** Our method enables the learning of parameters in NODEs without relying on numerical ODE solvers. Regarding computational efficiency, our approach requires only a series of global integrals for each trajectory. In these numerical integrals, the vector fields of NODEs are only evaluated at observed data points in parallel, significantly reducing the number of function evaluations and achieving better parallelizability. Additionally, our approach effectively mitigates error accumulation from autoregression, thus improving prediction accuracy.

## 5 EXPERIMENTS

In this section, we conduct a series of experiments to evaluate the performance of VF-NODEs. We begin by applying the method to four dynamical systems of various dimensions. Subsequently, we assess its performance on the real-world COVID-19 dataset. Lastly, we perform ablation studies to examine the model's key components.

**Baselines**. To evaluate acceleration performance, we compare our method against the following training approaches for NODE-based models: (1) the discretize-then-optimize approach (Dis-Opt), (2) the optimize-then-discretize approach (Opt-Dis) (Chen et al., 2018), and (3) the seminorm approach (Kidger et al., 2021). Additionally, we compare the *accuracy* of our proposed method against two categories of SOTA methods. The first category includes NODE-based models: (1) Vanilla NODE (Chen et al., 2018), (2) TayNODE (Kelly et al., 2020), (3) Latent ODE with RNN encoder (Chen et al., 2018), (4) ODE-RNN (Rubanova et al., 2019), (5) Latent ODE with ODE-RNN encoder (Rubanova et al., 2019), and (6) NCDE (Kidger et al., 2020). The second category consists of Neural Flows (Biloš et al., 2021), where neural networks $g_\theta$ are used to directly model the solution of ODEs as $x = g_\theta(t, x_0)$. This category includes models such as ResNet Flows and GRU Flows. Detailed hyperparameter settings of these models are provided in Appendix I.1. We also discuss other training methods for NODEs based on spline mtehods, as shown in Appendix H.

### 5.1 SIMULATION OF LOW-DIMENSIONAL DYNAMICAL SYSTEMS

**Dynamical Systems**. We select four dynamical systems from fields such as biology, biochemistry, genetics and epidemiology: the glycolytic oscillator (Sel'Kov, 1968), the genetic toggle switch (Gardner et al., 2000), the repressilator (Elowitz & Leibler, 2000), and the age-structured SIR model (Ram & Schaposnik, 2021). These systems vary in dimensionality: the repressilator is six-dimensional, and the age-structured SIR model is 27-dimensional, while the remaining systems are two-dimensional. Further dataset details are provided in Appendix I.2.

**Datasets**. For each system, except the age-structured SIR model, we generate 125 trajectories with randomly sampled initial conditions, of which 100 are used for training and 25 for validation. Each trajectory is uniformly sampled with 100 data points from $U[0, T]$, with values randomly dropped at a rate of $1 - r$ via masking. Following (Qian et al., 2022), Gaussian noise $\epsilon = \sigma_R \cdot \text{std}(x(t))$ is added to each observation. For baselines, each trajectory is segmented into 5 short segments to enhance the performance of baselines, while for VF-NODEs, the whole trajectory is used to improve the precision of spline regression. Additionally, we generate 25 test trajectories, each containing 200 randomly sampled points. The first 100 points are sampled from $U[0, T]$ for interpolation tasks, and the remaining 100 from $U[T, 2T]$ for extrapolation tasks. In our experiments, $T$ is set to 10, $\sigma_R$ to

0.01, and the retention ratio to $r = 0.8$. All settings remain the same for the age-structured SIR model except for the number of trajectories: 500 are generated for training and validation (400 for training and 100 for validation), and an additional 100 trajectories are generated for testing.

**Evaluation on Acceleration**. We compare the acceleration performance of our method against NODE-based baselines in terms of average training time per epoch using the glycolytic oscillator. As shown in Fig. 2, our method significantly accelerates the training of NODE-based baselines by a factor of 10 to 1000. This improvement is attributed to our approach, which relies solely on global numerical integration over the time domain, thereby reducing NFEs during training. Average training time per epoch on other dynamical systems are provided in Appendix G.1.

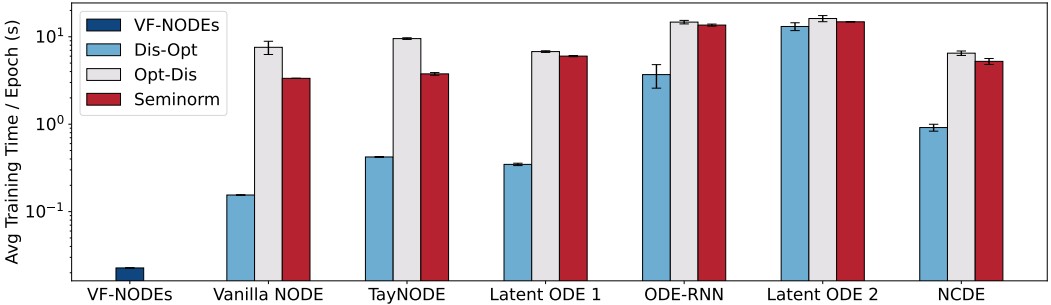

Figure 2: Average training time per epoch (second) for each method on the glycolytic model. Our method can achieve 10 to 1000 times faster than the baselines. Due to the high stability of the training speed for some methods, the uncertainty is negligible and not clearly visible in the figure.

**Evaluation on Prediction Error**. We evaluate the prediction error of different methods on interpolation and extrapolation tasks using mean squared error (MSE), as shown in Tables 1 and 2 respectively. The results demonstrate that our method consistently achieves higher or comparable performance compared wth the baselines. This superior performance can be attributed to two key factors: (1) the standard NODE architecture in VF-NODEs aligns well with dynamical systems represented by ODEs, and (2) the use of global integrals in the VF loss mitigates autoregression, effectively reducing error accumulation. Although Vanilla NODEs share the same neural architecture, they rely on ODE solvers for training, which leads to error accumulation and significantly slower training speeds compared to VF-NODEs. In the case of TayNODEs, the regularization term restricts the complexity of the learned dynamics, limiting the expressivity of the model and leading to even worse performance than Vanilla NODEs. Neural Flows also perform worse than VF-NODEs, as they must explicitly model the time dependencies of solutions. In contrast, VF-NODEs can leverage ODE solvers to handle time evolution during inference. Additionally, other baselines are designed for more general irregular time series data. They do not align with the form of these dynamical systems. Detailed training settings can be found in Appendix I.3.

Additionally, we test our method on the Gompertz model (Gompertz, 1825) and the Lotka-Volterra equations (Kingsland, 1995). Related experimental results can be found in Appendix G.2.

Table 1: Testing MSE (mean±standard deviation) for interpolation tasks on 4 dynamical systems with 80% observed data ($r = 0.8$). Lower values indicate better performance. Here e±n refers to $\times 10^{\pm n}$. Latent ODE 1 refers to Latent ODE with an RNN encoder. Latent ODE 2 refers to Latent ODE with an ODE-RNN encoder. The best results are highlighted in **bold black**, and the second-best results are highlighted in **bold purple**.

| | Glycolytic | Toggle | Repressilator | AgeSIR |
|---|---|---|---|---|
| Vanilla NODE | (1.51e-03)±(1.40e-03) | **(8.00e-04)±(8.69e-04)** | **(2.25e-02)±(5.04e-03)** | **(7.54e-03)±(6.58e-04)** |
| TayNODE | (3.20e-03)±(1.32e-03) | (1.37e-02)±(9.96e-03) | (1.43e-01)±(1.72e-02) | (3.18e-01)±(7.36e-02) |
| Latent ODE 1 | (2.21e-01)±(3.35e-02) | (6.57e-01)±(1.48e-01) | (2.33e+01)±(9.23e-01) | (4.18e+01)±(6.56e+00) |
| ODE-RNN | **(8.80e-05)±(2.30e-05)** | (1.63e-03)±(5.56e-04) | (1.41e-01)±(7.97e-03) | (6.75e+00)±(3.43e-01) |
| Latent ODE 2 | (1.00e-01)±(1.87e-02) | (6.31e-01)±(4.69e-01) | (7.47e-01)±(8.77e-02) | (1.36e+09)±(1.93e+09) |
| NCDE | (3.49e-02)±(1.36e-02) | (3.96e-02)±(3.43e-02) | (1.51e+00)±(1.22e+00) | (1.22e+01)±(3.75e+00) |
| ResNet Flow | (2.84e-01)±(5.69e-02) | (7.09e-01)±(2.21e-01) | (1.03e+01)±(8.14e-01) | (2.50e+00)±(1.96e-01) |
| GRU Flow | (3.80e-01)±(5.34e-02) | (2.45e+00)±(1.88e-01) | (7.45e+00)±(4.68e-02) | (4.19e+01)±(1.22e-01) |
| VF-NODE (Ours) | **(6.35e-05)±(2.68e-06)** | **(1.69e-04)±(6.09e-05)** | **(1.92e-02)±(2.62e-04)** | **(7.39e-03)±(6.71e-04)** |

Table 2: Testing MSE (mean±standard deviation) for extrapolation task on 4 dynamical systems with 80% observed data ($r = 0.8$). Lower values indicate better performance. Here e±n refers to $\times 10^{\pm n}$. Latent ODE 1 refers to Latent ODE with an RNN encoder. Latent ODE 2 refers to Latent ODE with an ODE-RNN encoder. The best results are highlighted in **bold black**, and the second-best results are highlighted in **bold purple**.

| | Glycolytic | Toggle | Repressilator | AgeSIR |
|---|---|---|---|---|
| Vanilla NODE | (8.79e-04)±(7.64e-04) | **(8.14e-07)±(6.72e-07)** | **(1.25e-01)±(3.11e-02)** | **(1.99e-02)±(1.69e-03)** |
| TayNODE | (4.71e-03)±(3.60e-03) | (5.09e-02)±(5.09e-02) | (8.73e-01)±(1.35e-01) | (4.52e-01)±(1.31e-01) |
| Latent ODE 1 | (2.39e-01)±(7.29e-02) | (1.48e+00)±(1.28e+00) | (9.14e+00)±(1.19e+00) | (2.30e+02)±(4.85e+01) |
| ODE-RNN | **(4.68e-05)±(1.15e-05)** | (2.04e-04)±(1.42e-04) | (2.02e-01)±(7.51e-03) | (7.48e+00)±(9.12e-02) |
| Latent ODE 2 | (1.82e-01)±(1.61e-01) | (4.96e+00)±(5.70e+00) | (3.09e+00)±(2.99e-01) | (1.36e+09)±(1.93e+09) |
| NCDE | (8.00e-01)±(4.48e-01) | (1.03e+00)±(7.38e-01) | (6.73e+00)±(4.85e+00) | (2.13e+01)±(8.27e+00) |
| ResNet Flow | (3.47e+00)±(2.82e+00) | (5.32e+00)±(1.97e+00) | (6.56e+01)±(2.23e+01) | (1.95e+01)±(3.29e-01) |
| GRU Flow | (7.39e-01)±(2.23e-01) | (5.03e+00)±(5.22e-01) | (1.84e+01)±(5.74e-01) | (6.02e+01)±(1.89e-01) |
| VF NODE (Ours) | **(1.63e-04)±(3.05e-05)** | **(4.79e-07)±(5.24e-08)** | **(1.23e-01)±(1.48e-02)** | **(2.37e-02)±(1.61e-03)** |

## 5.2 REAL-WORLD APPLICATION: COVID-19 DATASET

We also evaluate the performance of VF-NODEs on the real-world COVID-19 dataset. Using covsirphy (Takaya & Team, 2020–2024), we leverage data from the COVID-19 Data Hub (Guidotti & Ardia, 2020). This data can be modeled using a four-dimensional *phase-dependent* SIR-F framework (Takaya & Team, 2020–2024), where the ODE parameters vary over time. VF-NODEs, however, are not designed to handle these time-varying dynamics due to the limitations of the standard NODE architecture. To address this, we apply S-R analysis (Balkew, 2010) to segment the data, allowing the ODE parameters to remain constant within each segment. In this study, we evaluate our method and the baselines using data from four countries: Japan, Italy, Norway, and India. For each country, we extract 100 data points from the longest segment in a single trajectory, allocating 80 for training, 10 for validation, and 10 for testing. Each data point consists of four variables: Susceptible, Infected, Recovered, and Fatal. We standardize the data for consistency and employ spline regression to smooth it. The experimental results, presented in Table 3, demonstrate that our method achieves significantly better performance compared to the baselines. This superior performance can be attributed to the fact that the COVID-19 dataset conforms to the SIR-F ODE system, which aligns with the inductive bias of VF-NODEs. Vanilla NODEs outperform most other baselines because they share the same neural architecture as VF-NODEs. However, Vanilla NODEs have lower accuracy than our method due to the error accumulation caused by autoregression in ODE-solver-based training methods. Detailed training settings are provided in Appendix I.3.

Table 3: Testing MSE (mean±standard deviation) on the real-world COVID-19 dataset. Lower values indicate better performance. Here e±n refers to $\times 10^{\pm n}$. Latent ODE 1 refers to Latent ODE with an RNN encoder. Latent ODE 2 refers to Latent ODE with an ODE-RNN encoder. The best results are highlighted in **bold black**.

| | Japan | Italy | Norway | India |
|---|---|---|---|---|
| Vanilla NODE | (1.42e+00)±(7.44e-01) | (1.35e-02)±(1.39e-02) | (1.03e-03)±(6.30e-05) | (8.86e-04)±(3.76e-04) |
| TayNODE | (2.02e+00)±(8.00e-01) | (3.88e-02)±(5.70e-03) | (5.57e-04)±(1.13e-04) | (1.18e-02)±(1.08e-02) |
| Latent ODE 1 | (1.02e+01)±(4.42e-01) | (6.56e-01)±(2.87e-01) | (1.83e-01)±(1.10e-01) | (5.69e-01)±(1.59e-01) |
| ODE-RNN | (8.43e+00)±(6.56e-01) | (1.10e-01)±(1.12e-02) | (5.58e-03)±(1.48e-03) | (2.09e-01)±(1.74e-01) |
| Latent ODE 2 | (1.12e+01)±(1.31e+00) | (3.36e-01)±(2.70e-01) | (4.12e-01)±(2.90e-01) | (4.07e-01)±(1.54e-01) |
| NCDE | (1.14e+01)±(1.10e+00) | (8.72e-01)±(2.46e-01) | (1.27e-02)±(1.20e-02) | (3.94e-01)±(8.18e-02) |
| ResNet Flow | (9.33e-01)±(1.69e-01) | (3.69e-02)±(3.20e-03) | (2.51e-02)±(1.65e-02) | (2.04e-02)±(2.01e-03) |
| GRU Flow | (1.39e+00)±(2.96e-02) | (8.63e-03)±(1.17e-04) | (3.07e-03)±(3.07e-07) | (2.52e-03)±(1.84e-04) |
| VF NODE (Ours) | **(1.87e-01)±(4.82e-02)** | **(1.64e-03)±(2.19e-04)** | **(3.43e-04)±(1.18e-04)** | **(5.68e-04)±(2.28e-04)** |

Furthermore, we also apply the proposed VF-NODEs to model the temporal effects of chemotherapy on tumor volume. The experimental results are presented in Appendix G.3.

## 5.3 ABLATION STUDIES

Lastly, we conduct ablation studies to investigate the impact of various components on the performance of VF-NODEs. Specifically, we examine: (1) the influence of Filon's method and spline regression and (2) the type of basis functions. The experimental settings are consistent with those described in Section 5.1.

**Influence of Filon's method and spline regression**. There are two key components enabling the accurate computation of the VF loss in VF-NODEs: (1) Filon's method and (2) natural cubic spline regression. To evaluate the influence of these components, we assess the performance of VF-NODEs under the following settings: (1) using only Filon's method (Filon's method + natural cubic spline interpolation), (2) using only natural cubic spline regression (trapezoidal rule (Press, 2007) + natural cubic spline regression), (3) using neither (trapezoidal rule + natural cubic spline interpolation). (4) using Filon's method with Hermite cubic spline. The experimental results, shown in Table 4, demonstrate that while both components enhance performance, natural cubic spline regression has a more significant impact. The inferior performance of Hermite cubic spline can be attributed to its reliance on numerical differentiation, which is ill-posed on noisy, sparse data.

Table 4: Testing MSE (mean±standard deviation) for the ablation study on Filon's method and spline regression. Lower values indicate better performance. Here e±n refers to $\times 10^{\pm n}$.

| Filon | Spline | Regression | Glycolytic | | Toggle | |
|---|---|---|---|---|---|---|
| | | | Interpolation | Extrapolation | Interpolation | Extrapolation |
| ✓ | Natural | ✗ | (9.13e+05)±(1.29e+06) | (2.50e+30)±(3.53e+30) | (9.39e+16)±(1.33e+17) | (8.16e+37)±(1.15e+38) |
| ✗ | Natural | ✓ | (2.78e+01)±(2.65e+01) | (5.76e+02)±(5.67e+02) | (1.40e+09)±(1.40e+09) | (4.35e+20)±(4.35e+20) |
| ✗ | Natural | ✗ | (5.59e+01)±(5.48e+01) | (6.66e+01)±(6.55e+01) | (1.69e+14)±(1.69e+14) | (8.29e+29)±(8.29e+29) |
| ✓ | Hermite | ✗ | (1.54e+00)±(2.07e+00) | (3.90e+01)±(5.51e+01) | (3.74e-01)±(1.14e-01) | (9.66e-01)±(4.43e-01) |
| VF-NODE (Ours) | | | **(6.35e-05)±(2.68e-06)** | **(1.63e-04)±(3.05e-05)** | **(1.69e-04)±(6.09e-05)** | **(4.79e-07)±(5.24e-08)** |

**Type of basis functions**. To illustrate the rationale behind using Fourier basis and Filon's method, we test two alternative types of basis functions: (1) $\phi_\ell(t) = (t/T)(t/T - 1)(t/T - \ell/L)$, and (2) Hermite cubic spline interpolation at three points: $(0, 0)$, $(T/2, \ell/L)$, and $(T, 0)$, with derivatives at these points set to 0. Filon's method was not applied to these basis functions. The experimental results, shown in Table 5, indicate that VF-NODEs with Fourier basis significantly outperform those with other basis functions. This can be attributed to the requirement that $\phi_\ell$ should form a Hilbert orthonormal basis in the function space, as per Theorem 2. However, performing Gram-Schmidt orthonormalization on these basis functions is computationally expensive. Other Hilbert orthonormal basis functions, such as orthonormal polynomials Chihara (2011), also exist but fail to meet the boundary conditions $\phi_\ell(0) = \phi_\ell(T) = 0$. While Qian et al. (2022) used B-splines, their recursive generation (De Boor, 1978) can significantly slow down the training of VF-NODEs.

Table 5: Testing MSE (mean±standard deviation) for the ablation study on the types for basis functions $\phi_\ell$. Lower values indicate better performance. Here e±n refers to $\times 10^{\pm n}$.

| | Glycolytic | | Toggle | |
|---|---|---|---|---|
| | Interpolation | Extrapolation | Interpolation | Extrapolation |
| $(t/T)(t/T - 1)(t/T - \ell/L)$ | (3.89e-02)±(8.37e-03) | (1.56e-02)±(8.54e-03) | (3.44e-01)±(3.79e-01) | (6.23e-01)±(7.38e-01) |
| Hermite-based basis | (7.06e-01)±(5.39e-01) | (5.29e-01)±(4.99e-01) | (4.16e+00)±(4.55e+00) | (1.62e+01)±(2.06e+01) |
| VF-NODE (Ours) | **(6.35e-05)±(2.68e-06)** | **(1.63e-04)±(3.05e-05)** | **(1.69e-04)±(6.09e-05)** | **(4.79e-07)±(5.24e-08)** |

We also conduct some additional ablation studies, including **the sensitivity of VF-NODEs to hyperparameters**, **the effect of sampling**, and **the effect of noise**, which can be found in Appendix G.4.

## 6 CONCLUSION

This work introduced a novel training method based on variational formulation to accelerate the training of NODEs for dynamical systems. Our approach required only a series of global integrals in the loss computation, eliminating the need for traditional numerical ODE solvers. To overcome the challenges posed by oscillatory integrals in the VF loss, we incorporated Filon's method to enhance model performance. Additionally, we developed a natural cubic spline regression to better handle noisy and incomplete data. Extensive experiments demonstrated that our method significantly accelerates NODEs training while maintaining high accuracy. Limitations are discussed in Appendix A.

ACKNOWLEDGMENTS

Research reported in this paper was sponsored in part by NSF CPS 2311086, NSF CIRC 716152, and Faculty Research Grant at William & Mary 141446.

ETHICS STATEMENT

Our research is centered on methodological advancements in NODEs, specifically aiming to enhance the efficient modeling of dynamical systems in fields such as biology, epidemiology, and genetics. This work does not involve human subjects, personal data, or any form of invasive technology. All datasets used in our experiments are either generated from known ODEs or sourced from publicly available datasets, ensuring compliance with their respective licenses and usage policies.

REPRODUCIBILITY STATEMENT

We are committed to promoting reproducibility in scientific research. To facilitate this, we have provided comprehensive details of our experimental setups in Section 5 and Appendix I. This includes thorough descriptions of the datasets used, model architectures, hyperparameter settings, training procedures, and evaluation metrics. We will also release our source code for reproducing all experiments on a public repository upon publication.

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

## A  LIMITATIONS

A limitation of our approach is its reliance on polynomial-based approximation, which may not effectively approximate complex trajectories in intricate dynamical systems, such as chaotic systems, or may fail when the sampling is extremely sparse. However, the proposed VF-NODEs can still capture underlying patterns of chaotic systems, as shown in Appendix J. Additionally, computing the numerical integrals requires the trajectory points are available, meaning VF-NODEs are currently applicable only to time series tasks. In future work, we aim to explore new techniques to address these challenges, such as coordinate gradient descent (Matei et al., 2023).

## B  BACKGROUND: TRAINING OF NODEs

### B.1  FOUNDATIONAL TRAINING FRAMEWORKS OF NODEs

In current training frameworks of NODEs, the outcomes are directly computed using numerical ODE solvers for the forward pass. Consider a scalar-value loss function $\mathcal{L}(\cdot)$, the input is the result of an ODE solver:

$$\mathcal{L}(\boldsymbol{x}(t_1)) = \mathcal{L}\left(\boldsymbol{x}(t_0) + \int_{t_0}^{t_1} \boldsymbol{f_\theta}(t, \boldsymbol{x}) \mathrm{d}t\right) = \mathcal{L}\left(\mathrm{ODESolve}(\boldsymbol{f_\theta}, \boldsymbol{x}(t_0), t_0, t_1)\right), \quad (10)$$

Then for the backward pass, to compute gradients $\partial \mathcal{L}/\partial \boldsymbol{\theta}$, there are two major approaches: *discretize-then-optimize* and *optimize-then-discretize* (Kidger, 2022; Onken & Ruthotto, 2020). In the discretize-then-optimize approach, operations of ODE solvers are directly backpropagated. In contrast, the optimize-then-discretize approach, also known as the *adjoint sensitivity method*, introduces additional backward-in-time adjoint ODEs and calculates the gradients with an integral:

$$\frac{\mathrm{d}\boldsymbol{a}}{\mathrm{d}t} = -\boldsymbol{a}^\top \frac{\partial \boldsymbol{f_\theta}(t, \boldsymbol{x})}{\partial \boldsymbol{x}}, \quad \frac{\partial \mathcal{L}}{\partial \boldsymbol{\theta}} = \int_{t_1}^{t_0} \boldsymbol{a}^\top \frac{\partial \boldsymbol{f_\theta}(t, \boldsymbol{x})}{\partial \boldsymbol{\theta}} \mathrm{d}t. \quad (11)$$

where $\boldsymbol{a} = \frac{\partial \mathcal{L}}{\partial \boldsymbol{x}}$ is the *adjoint state*. Other training frameworks are typically modifications of these two approaches.

### B.2  ERROR ACCUMULATION OF NUMERICAL ODE SOLVERS

In this subsection, we discuss the issue of error accumulation in numerical ODE solvers. ODE solvers widely used in the context of NODEs can be expressed as

$$\boldsymbol{x}_{n+1} = \boldsymbol{x}_n + h\boldsymbol{g}(t_n, \boldsymbol{x}_n),$$

where $h$ is the step size, and $\boldsymbol{g}(\cdot, \cdot)$ denotes the updating formula. In essence, numerical ODE solvers use the state at the current time step to predict the state at the next time step, which can be viewed as a form of *autoregression*.

To demonstrate error accumulation, let us consider the simple ODE $\dot{x} = x$. The analytical solution of this ODE is $x(t) = x(0)\exp(t)$. We compute the mean absolute percentage error (MAPE) between the analytical solution and the numerical solution obtained using `LSODA` in `scipy` (Virtanen et al., 2020). The MAPE is shown in Fig. 3. As observed in the figure, the error increases over time. For untrained NODEs, such error accumulation can lead to poor performance during training.

## C  PROOF OF THEOREM 2

According to (Qian et al., 2022), we present the proof of Theorem 2, which is refined for neural networks.

*Proof of Theorem 2.* Given the definition of $\boldsymbol{c}$:

$$\boldsymbol{c}(\boldsymbol{x}, \boldsymbol{f_\theta}, \phi_\ell) := \int_0^T \boldsymbol{x}(t)\dot{\phi}_\ell(t) \, \mathrm{d}t + \int_0^T \boldsymbol{f_\theta}(t, \boldsymbol{x}(t))\phi_\ell(t) \, \mathrm{d}t. \quad (12)$$

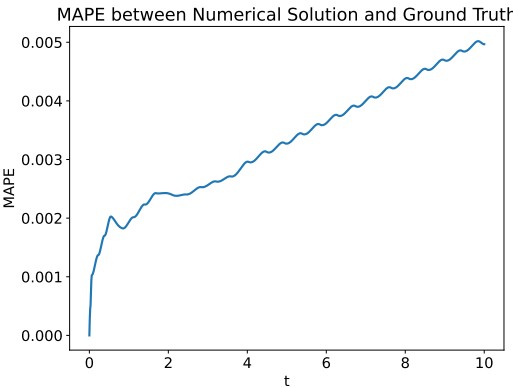

Figure 3: The MAPE between the numerical solution and ground truth (analytic solution) for $\dot{x} = x$.

For the first term in Eq. (12), we can express it as:

$$\int_0^T \boldsymbol{x}(t)\dot{\phi}_\ell(t)\,\mathrm{d}t = \int_0^T \boldsymbol{x}(t)\,\mathrm{d}\phi_\ell(t) = \boldsymbol{x}(T)\phi_\ell(T) - \boldsymbol{x}(0)\phi_\ell(0) - \int_0^T \dot{\boldsymbol{x}}(t)\phi_\ell(t)\,\mathrm{d}t$$
$$= -\int_0^T \dot{\boldsymbol{x}}(t)\phi_\ell(t)\,\mathrm{d}t\,. \tag{13}$$

Substituting Eq.(13) into Eq.(12), we obtain:

$$\boldsymbol{c}(\boldsymbol{x}, \boldsymbol{f_\theta}, \phi_\ell) = \int_0^T \boldsymbol{f_\theta}(t, \boldsymbol{x}(t))\phi_\ell(t)\,\mathrm{d}t - \int_0^T \dot{\boldsymbol{x}}(t)\phi_\ell(t)\,\mathrm{d}t$$
$$= \int_0^T \left(\boldsymbol{f_\theta}(t, \boldsymbol{x}(t)) - \dot{\boldsymbol{x}}(t)\right)\phi_\ell(t)\,\mathrm{d}t \tag{14}$$
$$= \int_0^T \left(\boldsymbol{f_\theta}(t, \boldsymbol{x}(t)) - \boldsymbol{f}(t, \boldsymbol{x}(t))\right)\phi_\ell(t)\,\mathrm{d}t\,.$$

Based on Parseval's identity, for each element in $\boldsymbol{c}(\boldsymbol{x}, \boldsymbol{f_\theta}, \phi_\ell)$, we can obtain:

$$\lim_{L\to\infty}\sum_{\ell=1}^L c_j^2(\boldsymbol{x}, \boldsymbol{f_\theta}, \phi_\ell) = \lim_{L\to\infty}\sum_{\ell=1}^L \int_0^T \left[\left(f_{\boldsymbol{\theta},j}(t, \boldsymbol{x}(t)) - f_j(t, \boldsymbol{x}(t))\right)\phi_\ell(t)\right]^2\,\mathrm{d}t$$
$$= \int_0^T \left[f_{\boldsymbol{\theta},j}(t, \boldsymbol{x}(t)) - f_j(t, \boldsymbol{x}(t))\right]^2\,\mathrm{d}t \tag{15}$$
$$= \|(f_{\boldsymbol{\theta},j} - f_j)\circ(t, \boldsymbol{x})\|_2^2$$

Thus, we have proven Theorem 2. $\qquad\square$

## D  GENERAL NUMERICAL INTEGRATION TECHNIQUES DO NOT WORK FOR OSCILLATORY INTEGRALS

For integrals over tabular data, Newton-Cotes formula-based methods (Press, 2007), such as the trapezoidal method and Simpson's method, are commonly used. These methods estimate the original integral by:

$$\int_0^T h(t)\sin(\omega t)\,\mathrm{d}t = \sum_{k=0}^{K-1}\int_{t_k}^{t_{k+1}} h(t)\sin(\omega t)\,\mathrm{d}t \approx \sum_{k=0}^{K-1}\int_{t_k}^{t_{k+1}} p_k(t)\,\mathrm{d}t\,, \tag{16}$$

where $p_k(t)$ is the approximate $n$-th-order polynomial of $h(t)\sin(\omega t)$ over the time interval $[t_k, t_{k+1}]$. This approach can compute the integral accurately when the original integrand is an exact

$n$-th-order polynomial in $[t_k, t_{k+1}]$. However, based on Taylor expansion, such polynomials only ensure accurate computation of an integral within an error bound of $O(g^{(n+1)})$ for any integrand $g(t)$ (Deaño et al., 2017). For the oscillatory integrand $h(t)\sin(\omega t)$, as $\omega$ increases, the $(n+1)$-th derivative of the integrand also increases rapidly and can be pretty large. In this scenario, polynomials may struggle to accurately approximate $h(t)\sin(\omega t)$, causing general numerical integration techniques to fail in calculating such integrals.

## E  SPLINE-BASED APPROXIMATION

In this section, we delve into the application of natural cubic splines for constructing closed-form approximations from sampled data. Generally, two techniques are employed for this purpose: (1) natural cubic spline interpolation, and (2) natural cubic spline regression.

Let's begin with natural cubic splines. Given sampled data $\{h(t_k)\}_{k=0}^{K}$, our objective is to construct a cubic polynomial $q_k(t)$ within each interval $[t_k, t_{k+1}]$:

$$q_k(t) = \sum_{m=0}^{3} a_{k,m}(t - t_k)^m = a_{k,0} + a_{k,1}(t - t_k) + a_{k,2}(t - t_k)^2 + a_{k,3}(t - t_k)^3, \qquad (17)$$

where $a_{k,m}$ $(k = 0, \ldots, K, \ m = 0, 1, 2, 3)$ are the coefficients of this polynomial. To ensure a smooth approximation in each interval, the following conditions must be met:

1. *Continuity*: The spline must be continuous at each point: $q_k(t_{k+1}) = q_{k+1}(t_{k+1})$.
2. *Smoothness*: The first and second derivatives of the spline must be continuous at each point: $q_k'(t_{k+1}) = q_{k+1}'(t_{k+1})$, and $q_k''(t_{k+1}) = q_{k+1}''(t_{k+1})$.
3. *Natural boundary condition*: The second derivative at the endpoints should be zero: $q_0''(t_0) = q_{K-1}''(t_K) = 0$.

Now, let's examine the difference between spline interpolation and spline regression. The crucial distinction lies in their treatment of the sampled data points. Spline interpolation constructs a piecewise cubic polynomial that exactly passes through all the given data points:

$$q_k(t_k) = h(t_k) \implies a_{k,0} = h(t_k).$$

However, it is not adept at handling noisy data as it strictly adheres to the given points. In contrast, spline regression constructs a piecewise cubic polynomial that best fits the data points in a least-squares sense while maintaining smoothness. This is achieved by solving the following optimization problem:

$$\min_{a_{k,m}} \left\{ \lambda \sum_{k=0}^{K} [h(t_k) - q(t_k)]^2 + (1 - \lambda) \int_0^T (q^{(2)}(t))^2 \, \mathrm{d}t \right\}, \qquad (18)$$

where $q(t) = q_k(t)$, $t \in [t_k, t_{k+1}]$, and $\lambda \in [0, 1]$ is a hyperparameter used to control smoothness. When $\lambda = 1$, this formulation reduces to spline interpolation. Spline regression is better suited for handling noisy data. The process of solving these coefficients $a_{k,m}$ can be found in (De Boor, 1978).

In our method, we initially employ spline regression to construct precise closed-form approximations of trajectories from noisy data, thereby obtaining estimated trajectories $\hat{x}(t)$. Subsequently, we utilize spline interpolation to construct accurate closed-form approximations of vector fields $f(t, \hat{x}(t))$.

## F  ALGORITHM OF VF-NODE

The detailed algorithm is presented in Algorithm 1. First, we employ natural cubic spline regression to construct an analytical approximation of trajectories from noisy and partially observed data $x$, represented as the coefficients of the spline. Next, we estimate the values in trajectories based on the spline to remove noise and fill in missing values. Then, we compute the vector fields $f$ based on estimated trajectories and utilize natural cubic spline regression again to build an analytical approximation of $f$. Finally, using Filon's method, we compute oscillatory integrals based on the analytical approximations of $x$ and $f$.

---

**Algorithm 1:** VF Loss

---

**Data:** A trajectory $\{(t_k, \boldsymbol{x}(t_k))\}_{k=0}^{K}$, where $0 = t_0 < t_1 < \cdots < t_K = T$, and
$\quad \boldsymbol{x}(t_k) = [x_1(t_k), x_2(t_k), \ldots, x_d(t_k)]^\top \in (\mathbb{R} \cup \{*\})^d$

**Input:** Smoothing coefficient $\lambda \in [0, 1]$, the number of basis functions $L$, and neural network
$\quad \boldsymbol{f_\theta}(\cdot, \boldsymbol{x}(\cdot))$

**Output:** The VF loss

/\*Perform spline regression on $\boldsymbol{x}$ to get spline coefficients \*/

$\boldsymbol{a}_{k,m} = \text{SplineRegression}(\lambda, \{(t_k, \boldsymbol{x}(t_k))\}_{k=0}^{K}) \in \mathbb{R}^d, \ k = 0, \ldots, K-1, \ m = 0,1,2,3$

/\*Make estimations of the trajectory $\boldsymbol{x}$ \*/

**for** $k = 0 : K$ **do**
     **if** $k < K$ **then**
         $\hat{\boldsymbol{x}}(t_k) = \boldsymbol{a}_{k,0} + \sum_{m=1}^{3} \boldsymbol{a}_{k,m}(t_k - t_k)^m = \boldsymbol{a}_{k,0}$
     **else**
         $\hat{\boldsymbol{x}}(t_K) = \sum_{m=0}^{3} \boldsymbol{a}_{K-1,m}(t_K - t_{K-1})^m$

/\*Evaluate the vector fields $\boldsymbol{f}$ \*/

$\boldsymbol{f}(t_k) = \boldsymbol{f_\theta}(t_k, \hat{\boldsymbol{x}}(t_k)), \ k = 0, \ldots, K$

/\*Perform spline interpolation on $\boldsymbol{f}$ to get spline coefficients \*/

$\boldsymbol{b}_{k,m} = \text{SplineInterp}(\{(t_k, \boldsymbol{f}(t_k))\}_{k=0}^{K}) \in \mathbb{R}^d, \ k = 0, \ldots, K-1, \ m = 0,1,2,3$

/\*Compute oscillatory integrals \*/

**for** $\ell = 1 : L$ **do**
     $\int_0^T \hat{\boldsymbol{x}}(t) \dot{\phi}_\ell(t)\, \mathrm{d}t = \sqrt{\frac{2}{T}} \frac{\pi \ell}{T} \sum_{k=0}^{K-1} \sum_{m=0}^{3} \boldsymbol{a}_{k,m} \int_{t_k}^{t_{k+1}} (t - t_k)^m \cos \frac{\pi \ell t}{T}\, \mathrm{d}t$
     $\int_0^T \boldsymbol{f_\theta}(t, \hat{\boldsymbol{x}}(t)) \phi_\ell(t)\, \mathrm{d}t \approx \sqrt{\frac{2}{T}} \sum_{k=0}^{K-1} \sum_{m=0}^{3} \boldsymbol{b}_{k,m} \int_{t_k}^{t_{k+1}} (t - t_k)^m \sin \frac{\pi \ell t}{T}\, \mathrm{d}t$
     $\boldsymbol{c}(\boldsymbol{x}, \boldsymbol{f}, \phi_\ell) = \int_0^T \boldsymbol{f_\theta}(t, \hat{\boldsymbol{x}}(t)) \phi_\ell(t)\, \mathrm{d}t + \int_0^T \hat{\boldsymbol{x}}(t) \dot{\phi}_\ell(t)\, \mathrm{d}t$

**return** $\sum_{\ell=1}^{L} \|\boldsymbol{c}(\boldsymbol{x}, \boldsymbol{f}, \phi_\ell)\|_2^2$

---

## G  ADDITIONAL EXPERIMENTS

### G.1  ADDITIONAL RESULTS: AVERAGE TRAINING TIME ON OTHER DYNAMICAL SYSTEMS

In addition to the average training time per epoch on the glycolytic model presented in Fig. 2, we also provide additional timing results for the repressilator model and the age-structured SIR model, as shown in Fig. 4 and Fig. 5, respectively. For some methods, the uncertainty may not be visible due to their stability. Across different cases, the proposed VF-NODEs achieve acceleration factors ranging from 10 to 1000 times compared to the baselines.

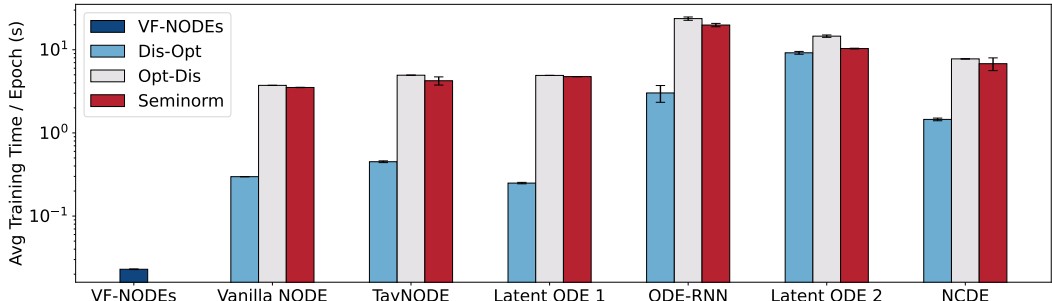

Figure 4: Average training time per epoch (second) for each method on the repressilator model. Our method can achieve 10 to 1000 times faster than the baselines. Due to the high stability of the training speed for some methods, the uncertainty is negligible and not clearly visible in the figure.

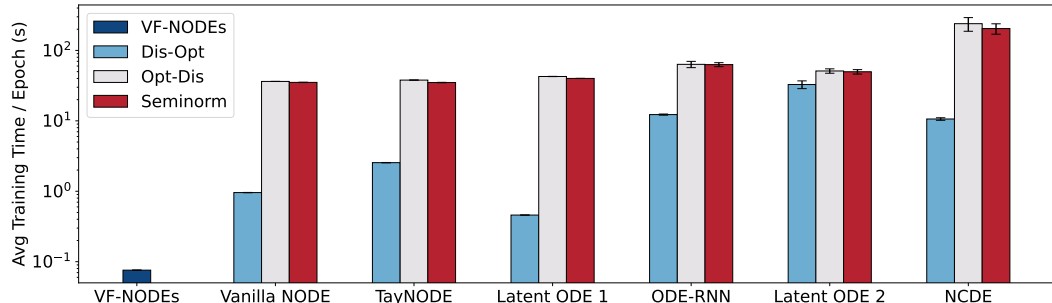

Figure 5: Average training time per epoch (second) for each method on the age-structured SIR model. Our method can achieve 10 to 1000 times faster than the baselines. Due to the high stability of the training speed for some methods, the uncertainty is negligible and not clearly visible in the figure.

### G.2  ADDITIONAL EXPERIMENTS ON SIMULATION FOR DYNAMICAL SYSTEMS

Besides four dynamical systems mentioned in Section 5.1, we also test two additional dynamical systems: (1) the Gompertz model (Gompertz, 1825) and (2) Lotka-Volterra equations (Kingsland, 1995). The Testing MSE for our method and other baselines are presented in Table 6. These additional experimental results also demonstrate the outstanding performance of the proposed VF-NODEs. Detailed training settings are presented in Appendix I.3.

### G.3  TEMPORAL EFFECT OF CHEMOTHERAPY ON TUMOR VOLUME

We also use VF-NODEs to model the temporal effect of chemotherapy on tumor volume. Using the same data preprocessing as (Qian et al., 2022), the results are shown in Table 7. NCDEs failed due to the setting in diffrax (Kidger, 2022). Note that our method does not outperform all baselines due to the inherent limitations of the standard NODE architecture, which assumes that time series data must be the solution of an ODE system. In contrast, ODE-RNNs, which combine ODEs with RNNs, are not bound by this constraint. However, it is important to note that the training of VF-NODEs can be almost 180 times faster than that of ODE-RNNs. Detailed training settings are provided in Appendix I.3.

Table 6: Testing MSE (mean±standard deviation) on two additional dynamical systems with 80% observed data ($r = 0.8$). Lower values indicate better performance. Here e±n refers to $\times 10^{\pm n}$. Latent ODE 1 refers to Latent ODE with an RNN encoder. Latent ODE 2 refers to Latent ODE with an ODE-RNN encoder. The best results are highlighted in **bold black**, and the second-best results are highlighted in **bold purple**.

| | Gompertz | | Lotka-Volterra | |
|---|---|---|---|---|
| | Interpolation | Extrapolation | Interpolation | Extrapolation |
| Vanilla NODE | **(3.18e-07)±(9.26e-08)** | **(1.76e-07)±(8.74e-10)** | **(1.21e+00)±(5.99e-01)** | **(2.11e+01)±(5.89e+00)** |
| TayNODE | (5.15e-05)±(1.02e-05) | (4.12e-06)±(2.64e-06) | (1.50e+02)±(8.24e+00) | (1.64e+02)±(2.85e+00) |
| Latent ODE 1 | (4.54e-03)±(3.56e-04) | (2.91e-03)±(3.87e-03) | (4.62e+02)±(9.22e+01) | (1.69e+03)±(7.16e+02) |
| ODE-RNN | (1.52e-05)±(1.42e-05) | (2.04e-06)±(1.82e-06) | (1.97e+01)±(2.50e+01) | (1.32e+02)±(1.84e+02) |
| Latent ODE 2 | (4.37e-03)±(4.78e-05) | (2.26e-03)±(2.18e-04) | (2.39e+03)±(1.85e+03) | (1.35e+03)±(1.34e+02) |
| NCDE | (5.37e-04)±(2.92e-04) | (3.50e-02)±(4.10e-02) | (3.94e+00)±(2.52e+00) | (3.14e+01)±(2.47e+01) |
| ResNet Flow | (1.81e-02)±(1.86e-02) | (5.63e+00)±(7.79e+00) | (2.57e+02)±(9.53e+01) | (2.98e+02)±(3.68e+01) |
| GRU Flow | (2.27e-03)±(1.63e-03) | (4.95e-02)±(3.62e-02) | (3.69e+02)±(1.15e-01) | (4.83e+02)±(9.83e+00) |
| VF-NODE (Ours) | **(2.76e-07)±(1.70e-07)** | **(1.78e-07)±(4.49e-09)** | **(1.00e+00)±(3.01e-03)** | **(1.36e+01)±(4.66e+00)** |

Table 7: Testing RMSE (mean±standard deviation) of modeling the temporal effect of chemotherapy on tumor volume. Lower values indicate better performance. Here e±n refers to $\times 10^{\pm n}$. The best results are highlighted in **bold black**, and the second-best results are highlighted in **bold purple**.

| | RMSE |
|---|---|
| Vanilla NODE | (1.89e-03)±(9.76e-06) |
| TayNODE | (2.31e-03)±(1.43e-05) |
| Latent ODE (RNN Enc.) | (3.88e-03)±(1.27e-04) |
| ODE-RNN | **(1.59e-03)±(3.43e-04)** |
| Latent ODE (ODE-RNN Enc.) | (9.02e-03)±(4.42e-03) |
| NCDE | (nan)±(nan) |
| ResNet Flow | (1.88e-03)±(3.46e-06) |
| GRU Flow | (3.14e-03)±(3.66e-06) |
| VF-NODE (Ours) | **(1.87e-03)±(3.71e-05)** |

## G.4 ADDITIONAL EXPERIMENTS ON ABLATION STUDIES

In this subsection, we first evaluate the sensitivity of the proposed VF-NODEs to two hyper-parameters: the number of basis functions $L$ and spline regression hyper-parameter $\lambda$ mentioned in Appendix E. Subsequently, we assess the performance of VF-NODEs under varying sampling and noise conditions.

**The number of basis functions** $L$. We set the number of basis function $L$ to test the impact of $L$ on the performance and training speed of VF-NODEs by setting $L = 50, 60, 70, 80, 90, 100, 110$, as shown in Fig. 6 and Fig. 7, respectively. We can see that when $L$ increases, the testing MSE decreases generally. This can be attributed to Filon's method and spline regression, which enable us to compute oscillatory integrals precisely from noisy and partially observed data. Additionally, Fig. 7 demonstrate that the training speed of VF-NODEs is not sensitive to $L$. Considering the computational cost, we set $L = 80$ in all experiments.

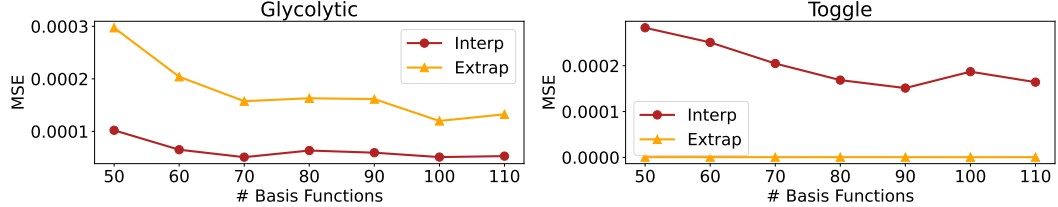

Figure 6: The impact of the number of basis functions $L$ in the VF loss on model performance

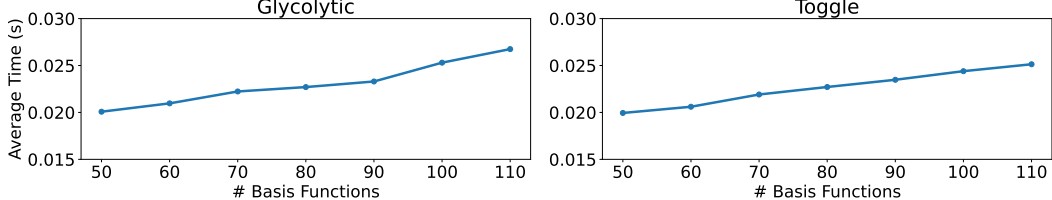

Figure 7: The impact of the number of basis functions $L$ in the VF loss on training speed of VF-NODEs.

**Spline Regression Hyper-parameter** $\lambda$. As discussed in Appendix E, $\lambda$ controls the smoothness of the splines. Typically, $\lambda$ values greater than 0.9 are suitable for most cases. To evaluate the sensitivity of the proposed VF-NODEs to $\lambda$, we test VF-NODEs with $\lambda = 0.9, 0.99, 0.9999$, and $0.99999$. The experimental results are presented in Table 8. These results indicate that our method is robust to variations in $\lambda$. In practice, $\lambda$ can be selected based on the noise level of the sampled trajectories, with higher noise levels generally requiring smaller values of $\lambda$, making it a straightforward parameter to adjust.

Table 8: Testing MSE (mean±standard deviation) from the ablation study on the sensitivity of VF-NODEs to $\lambda$. Lower values indicate better performance. Here e±n refers to $\times 10^{\pm n}$.

| $\lambda$ | Glycolytic | | Toggle | |
|---|---|---|---|---|
| | Interpolation | Extrapolation | Interpolation | Extrapolation |
| 0.9 | (4.20e-03)±(3.44e-04) | (4.81e-03)±(6.03e-04) | (5.40e-03)±(2.46e-04) | (7.57e-06)±(5.49e-06) |
| 0.99 | (1.14e-03)±(2.00e-04) | (1.78e-03)±(7.26e-04) | (2.16e-03)±(7.42e-04) | (8.29e-06)±(9.32e-06) |
| 0.999 | (7.27e-04)±(6.59e-04) | (7.80e-04)±(5.77e-04) | (1.79e-03)±(9.64e-04) | (1.52e-05)±(9.32e-06) |
| 0.9999 | (5.32e-04)±(4.47e-04) | (1.41e-03)±(1.59e-03) | (8.94e-03)±(1.09e-02) | (3.40e-02)±(4.80e-02) |
| 0.99999 | (6.35e-05)±(2.68e-06) | (1.63e-04)±(3.05e-05) | (1.69e-04)±(6.09e-05) | (4.79e-07)±(5.24e-08) |

**Effect of sampling**. There are three key settings related to sampling: (1) the number of data points per trajectory, (2) the random dropping rate, $1 - r$, and (3) sampling interval $T$. First, we fix $r = 0.8$,

$T = 10$ and sample 50, 100, 300, 500, 700, and 1000 points for each trajectory, respectively. The experimental results are presented in Table 9. Next, we fix the number of points at 100 and test with $r = 0.4$, 0.6, and 0.8, as presented in Table 10. Finally, we vary the sampling interval, setting $T = 10$, 20, and 50, while considering 100, 200, 500 sampled points respectively, as reported in Table 11. These experiments demonstrate that our method is robust across different sampling settings, particularly on sparsely sampled data. This can be attributed to spline regression's ability to perform effectively on sparse datasets (Knowles & Renka, 2014).

Table 9: Testing MSE (mean±standard deviation) from the ablation study on the number of sampling points for each trajectory. Lower values indicate better performance. Here e±n refers to $\times 10^{\pm n}$.

| # Point | Glycolytic | | Toggle | |
| --- | --- | --- | --- | --- |
| | Interpolation | Extrapolation | Interpolation | Extrapolation |
| 50 | (5.28e-04)±(3.23e-04) | (7.08e-04)±(3.36e-04) | (1.56e-03)±(8.33e-04) | (4.56e-06)±(8.45e-07) |
| 100 | (6.35e-05)±(2.68e-06) | (1.63e-04)±(3.05e-05) | (1.69e-04)±(6.09e-05) | (4.79e-07)±(5.24e-08) |
| 300 | (4.90e-05)±(1.74e-05) | (1.20e-04)±(3.72e-05) | (2.09e-04)±(7.24e-05) | (1.73e-07)±(4.65e-08) |
| 500 | (4.97e-05)±(1.39e-05) | (1.01e-04)±(3.11e-05) | (2.02e-04)±(1.00e-04) | (2.21e-07)±(7.07e-08) |
| 700 | (4.46e-05)±(1.39e-05) | (8.39e-05)±(1.17e-05) | (1.28e-04)±(5.15e-05) | (1.05e-07)±(5.15e-08) |

Table 10: Testing MSE (mean±standard deviation) from the ablation study for the random dropping rate $1 - r$. Lower values indicate better performance. Here e±n refers to $\times 10^{\pm n}$.

| $r$ | Glycolytic | | Toggle | |
| --- | --- | --- | --- | --- |
| | Interpolation | Extrapolation | Interpolation | Extrapolation |
| 0.4 | (1.05e-04)±(2.27e-05) | (2.46e-04)±(1.42e-04) | (5.84e-04)±(1.19e-04) | (1.51e-06)±(4.24e-07) |
| 0.6 | (8.09e-05)±(4.52e-05) | (1.72e-04)±(8.13e-05) | (3.75e-04)±(6.26e-05) | (7.23e-07)±(2.31e-07) |
| 0.8 | (6.35e-05)±(2.68e-06) | (1.63e-04)±(3.05e-05) | (1.69e-04)±(6.09e-05) | (4.79e-07)±(5.24e-08) |

Table 11: Testing MSE (mean±standard deviation) from the ablation study on the sampling interval $T$ for each trajectory. Lower values indicate better performance. Here e±n refers to $\times 10^{\pm n}$.

| $T$ | Glycolytic | | Toggle | |
| --- | --- | --- | --- | --- |
| | Interpolation | Extrapolation | Interpolation | Extrapolation |
| 10 | (6.35e-05)±(2.68e-06) | (1.63e-04)±(3.05e-05) | (1.69e-04)±(6.09e-05) | (4.79e-07)±(5.24e-08) |
| 20 | (4.11e-04)±(4.60e-04) | (9.27e-04)±(8.23e-04) | (1.24e-04)±(6.65e-05) | (2.28e-07)±(4.25e-08) |
| 50 | (3.22e-04)±(1.21e-05) | (2.34e-03)±(9.19e-04) | (3.86e-04)±(3.18e-04) | (1.90e-07)±(8.22e-08) |

**Effect of noise**. We set the relative noise level $\sigma_R = 0.01$, 0.03, 0.05, 0.07, 0.1, 0.3, and 0.5 respectively. The experimental results are shown in Fig. 8. By setting the smoothing hyperparameter $\lambda = 0.99$ in spline regression (as discussed in Appendix E), VF-NODEs demonstrate robustness to different noise settings.

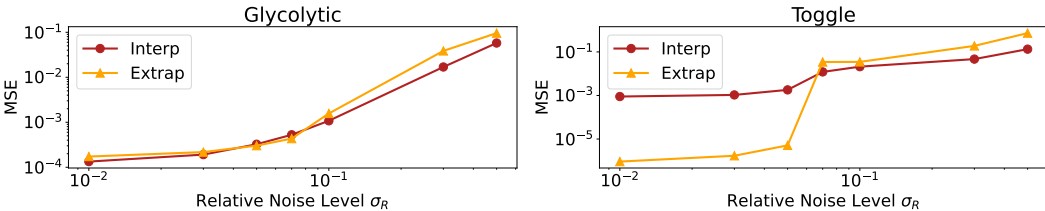

Figure 8: Testing MSE on the effect of relative noise level $\sigma_R$.

## H    DISCUSSION: TRAINING NODEs BASED ON NATURAL CUBIC SPLINES

The proposed VF-NODEs utilize natural cubic splines to numerically compute integrals in the VF loss. Beyond this, some other training methods for NODEs also leverage splines. One such method is gradient matching (Varah, 1982; Li et al., 2024), where the derivatives of sampled trajectories are estimated numerically using splines or other spectral methods. The DNN-based vector fields are then trained to match these estimated derivatives. In the second method, given the sampled trajectories $\{\boldsymbol{x}(t_k)\}_{k=0}^{K}$, the DNN-based vector fields can be approximated as

$$\boldsymbol{f_\theta}(t, \hat{\boldsymbol{x}}(t)) \approx \boldsymbol{b}_{k,0} + \sum_{m=1}^{3} \boldsymbol{b}_{k,m}(t - t_k)^m, \ t \in [t_k, t_{k+1}]. \tag{19}$$

Using this approximation, the solution of NODEs at the sampled time points can be computed as

$$
\begin{aligned}
\hat{\boldsymbol{x}}(t_{k+1}) &= \hat{\boldsymbol{x}}(t_k) + \int_{t_k}^{t_{k+1}} \boldsymbol{f_\theta}(\tau, \hat{\boldsymbol{x}}(\tau))\, \mathrm{d}\tau \\
&\approx \hat{\boldsymbol{x}}(t_k) + \int_{t_k}^{t_{k+1}} \left[\boldsymbol{b}_{k,0} + \sum_{m=1}^{3} \boldsymbol{b}_{k,m}(\tau - t_k)^m\right] \mathrm{d}\tau \\
&= \hat{\boldsymbol{x}}(t_k) + \sum_{m=0}^{3} \boldsymbol{b}_{k,m} \frac{(t_{k+1} - t_k)^{m+1}}{m + 1}.
\end{aligned}
\tag{20}
$$

The NODE can then be trained by minimizing the MSE between $\boldsymbol{x}(t_k)$ and $\hat{\boldsymbol{x}}(t_k)$. All these methods are ODE-solver-free, similar to the proposed VF-NODEs.

We refer to these methods as *Grad-Matching NODEs* and *Spline-Integ NODEs*, respectively. We compared the proposed VF-NODEs with these methods, with experimental results presented in Tables 12 and 13. While the training speed of these methods is similar (approximately $10^{-2}$ second per epoch), we observe that their performance is significantly worse than that of VF-NODEs.

For Grad-Matching NODEs, this performance gap can be attributed to their sensitivity to noisy and sparse sampling. In contrast, VF-NODEs rely solely on numerical integrals, representing an improvement and generalization of gradient matching methods Brunel et al. (2014).

For Spline-Integ NODEs, the discrepancy arises due to the autoregressive nature as described in Eq. (20). Errors accumulate across iterations because each step depends on the approximation from the previous one. VF-NODEs, by relying on global numerical integrals, avoid this issue of error accumulation, resulting in more robust and accurate training.

Table 12: Testing MSE (mean±standard deviation) for interpolation task of NODEs trained with splines on 4 dynamical systems with 80% observed data ($r = 0.8$). Lower values indicate better performance. Here e±n refers to $\times 10^{\pm n}$. The best results are highlighted in **bold black**.

|  | Glycolytic | Toggle | Repressilator | AgeSIR |
|---|---|---|---|---|
| Grad-Matching NODE | (7.91e-04)±(6.52e-04) | (1.19e-03)±(8.55e-04) | (5.48e-02)±(9.20e-03) | (9.50e-03)±(2.08e-04) |
| Spline-Integ NODE | (1.34e-03)±(1.44e-03) | (9.58e-03)±(7.76e-03) | (9.64e-02)±(1.01e-02) | (3.48e-02)±(2.34e-03) |
| VF NODE (Ours) | **(6.35e-05)±(2.68e-06)** | **(1.69e-04)±(6.09e-05)** | **(1.92e-02)±(2.62e-04)** | **(7.39e-03)±(6.71e-04)** |

Table 13: Testing MSE (mean±standard deviation) for extrapolation task of NODEs trained with splines on 4 dynamical systems with 80% observed data ($r = 0.8$). Lower values indicate better performance. Here e±n refers to $\times 10^{\pm n}$. The best results are highlighted in **bold black**.

|  | Glycolytic | Toggle | Repressilator | AgeSIR |
|---|---|---|---|---|
| Grad-Matching NODE | (6.32e-04)±(1.86e-04) | (4.43e-06)±(4.22e-06) | (3.87e-01)±(4.90e-02) | (3.16e-02)±(1.38e-03) |
| Spline-Integ NODE | (5.44e-03)±(6.45e-03) | (5.22e-05)±(3.55e-05) | (6.05e-01)±(7.54e-02) | (1.68e-01)±(8.96e-03) |
| VF NODE (Ours) | **(1.63e-04)±(3.05e-05)** | **(4.79e-07)±(5.24e-08)** | **(1.23e-01)±(1.48e-02)** | **(2.37e-02)±(1.61e-03)** |

## I  EXPERIMENTAL SETTINGS

All experiments in this work are implemented using `jax` (Bradbury et al., 2018). Specifically, the implementation of neural differential equation models is based on `equinox` (Kidger & Garcia, 2021) and `diffrax` (Kidger, 2022). To optimize models, we use the `optax` (DeepMind et al., 2020). All the experiments are implemented on the same server, equipped with 4 A5000 GPUs with 24GB graphics memory.

### I.1  HYPERPARAMETERS FOR MODELS

To evaluate the performance of VF-NODEs, we compare them with two categories of baseline models: NODE-based models and Neural Flows. To maintain consistent parameters across all models, we employed the following hyperparameters:

- VF-NODEs: the vector field is parameterized as a 4-layer MLP with 128 hidden units per layer. For the VF loss:
  - The number of basis functions $L$ is set to 80 for all tasks.
  - the smoothing hyperparameter $\lambda$ is set to 0.99999 for most tasks, while for the simulation of the age-structured SIR model, $\lambda$ is set to 0.9999.
- Vanilla NODEs (Chen et al., 2018): the vector field is parameterized as a 4-layer MLP with 128 hidden units per layer.
- TayNODEs (Kelly et al., 2020):
  - The neural architecture of TayNODEs is the same as that of Vanilla NODEs.
  - The 5-th order derivative regularization is used to match the order of `Dopri5` solver. $\lambda$ is set to 0.001.
- Latent ODEs with RNN encoders (Chen et al., 2018):
  - RNN encoder: a 1-layer GRU with 25 hidden units.
  - NODE Decoder: the vector field is parameterized as a 4-layer MLP with 124 hidden units per layer.
  - The latent size is set to 4.
- ODE-RNNs Rubanova et al. (2019): 1-layer GRU with 4 hidden units + 4-layer MLP with 127 hidden units per layer for NODE.
- Latent ODEs with ODE-RNN encoders (Rubanova et al., 2019):
  - ODE-RNN Encoder: 1-layer GRU with 4 hidden units and a 4-layer MLP with 84 hidden units per layer for NODE.
  - NODE Decoder: A 4-layer MLP with 84 hidden units per layer for NODE.
  - Latent size is set to 4.
- NCDEs (Kidger et al., 2020): The hidden size is set to 4. A 4-layer MLP with 89 hidden units per layer is used for the vector field.
- ResNet Flow (Biloš et al., 2021): 4-layer MLP with 128 hidden units per layer.
- GRU Flow (Biloš et al., 2021): Three 4-layer MLP with 74 hidden units per layer.

In most experiments, we used the `Dopri5` solver for NODE-based models. However, because training ODE-RNNs is computationally intensive, we employed the `Midpoint` solver for these models to reduce computational complexity.

### I.2  SETTINGS FOR DYNAMICAL SYSTEMS

In this subsection, we provide detailed information on the dynamical systems used in this work, including their trajectories and the simulation results of VF-NODEs compared to the best baseline, as reported in Tables 1 and 2. For this study, we selected six dynamical systems. The specific parameters and trajectories for each system are detailed below.

**Glycolytic oscillator** (Sel'Kov, 1968). The glycolytic oscillator is a fundamental system in biochemistry that models the glycolysis process. It can be expressed as

$$\dot{x}_1 = \theta_1 - \theta_2 x_1 - x_1 x_2^2,$$
$$\dot{x}_2 = -x_2 + \theta_3 x_1 + x_1 x_2^2,$$

(21)

where $\theta_1 = 0.75$, $\theta_2 = \theta_3 = 0.1$, and $x_1(0), x_2(0) \in [0.1, 1.1]$.

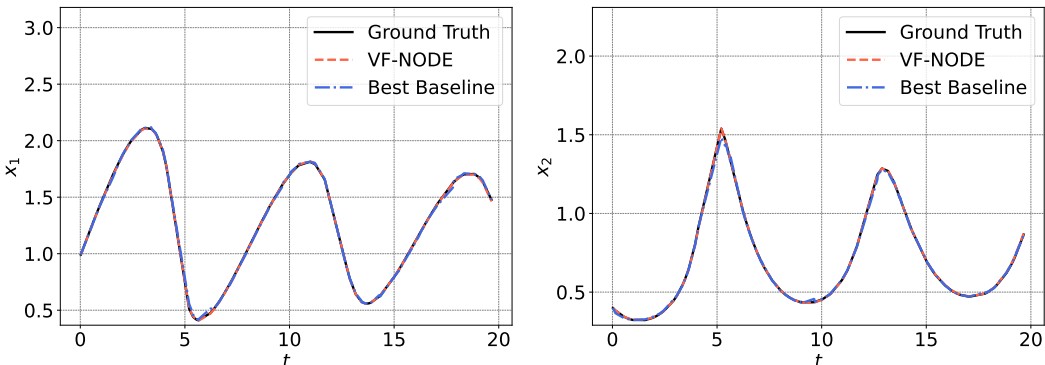

Figure 9: The trajectory plots of the glycolytic model with simulation results of VF-NODEs and the best baseline (ODE-RNN).

**Genetic toggle switch** (Gardner et al., 2000). The genetic toggle switch is a key mechanism in genetic engineering and synthetic biology for controlling genes. It can be expressed as

$$\dot{x}_1 = \frac{a_1}{1 + x_2^{n_1}} - x_1,$$
$$\dot{x}_2 = \frac{a_2}{1 + x_1^{n_2}} - x_2,$$

(22)

where $a_1 = a_2 = 4$, $n_1 = n_2 = 3$, and $x_1(0), x_2(0) \in [0.1, 4.0]$.

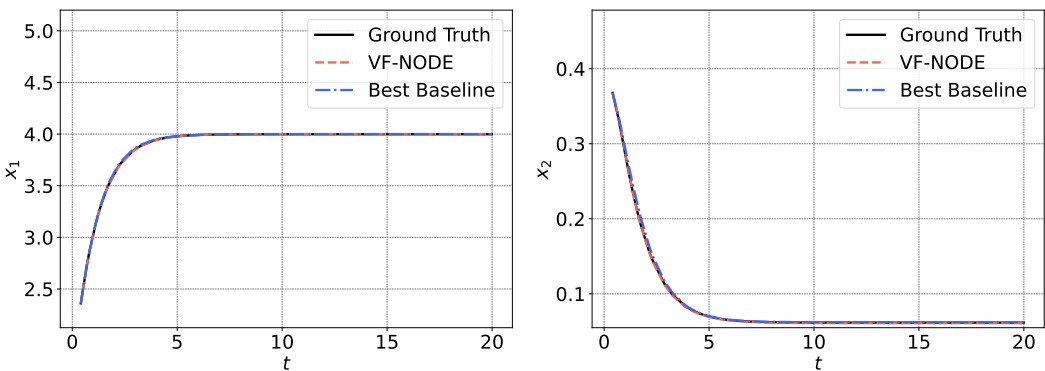

Figure 10: The trajectory plots of the toggle switch model with simulation results of VF-NODEs and the best baseline (Vanilla NODE).

**Repressilator** (Elowitz & Leibler, 2000). The repressilator is a genetic regulatory network. It can be expressed as

$$\dot{m}_i = -m_i + \frac{\alpha}{1 + \rho_j^n} + \alpha_0, \quad i = \text{lacI}, \text{tetR}, \text{cI},$$
$$\dot{\rho}_i = -\beta(\rho_i - m_i), \qquad j = \text{cI}, \text{lacI}, \text{tetR},$$

(23)

where $\rho_i$ are three repressor-protein concentrations, and $m_i$ are corresponding mRNA concentrations. We set $\alpha = 10$, $\alpha_0 = 10^{-5}$, $\beta = 1$, $n = 3$, and $m_i(0), \rho_i(0) \in [0, 5]$.

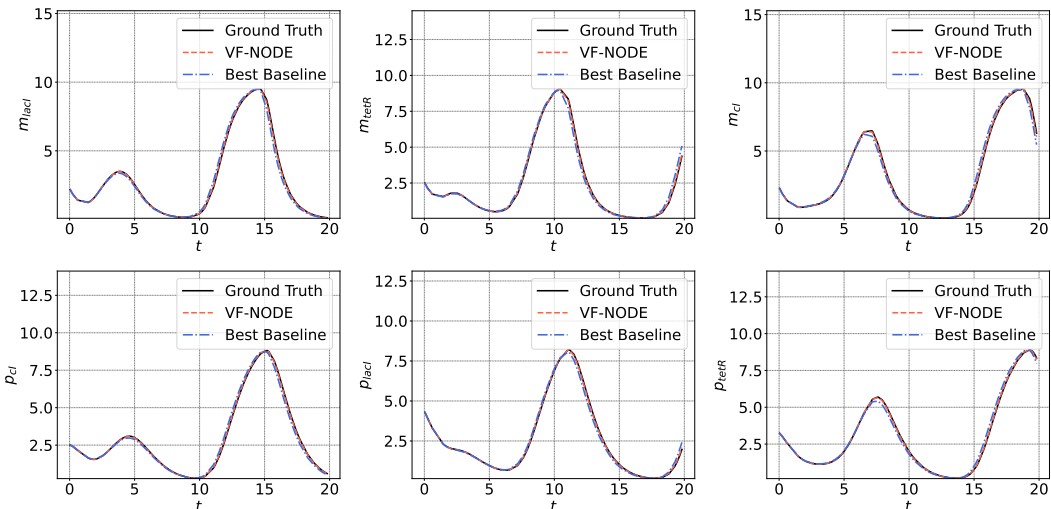

Figure 11: The trajectory plots of the repressilator model with simulation results of VF-NODEs and the best baseline (Vanilla NODE).

**Age-structured SIR model** (Ram & Schaposnik, 2021). The age-structured SIR model is a variant of the standard SIR model that considers the effects of different age groups. It can be expressed as

$$
\dot{S}_i = -\beta \frac{S_i}{N} \sum_{j=1}^{n} M_{ij} I_j,
$$
$$
\dot{I}_i = \beta \frac{S_i}{N} \sum_{j=1}^{n} M_{ij} I_j - \gamma I_i, \tag{24}
$$
$$
\dot{R}_i = \gamma I_i,
$$

where $S_i$, $I_i$, and $R_i$ ($i = 1, \ldots, 9$) denote the numbers of susceptible, infected, and removed individuals, respectively, for the age groups 0–9, 10–19, ..., 70–79, and 80+. The age-contact matrix $M$ is parameterized as

$$
M = \begin{bmatrix}
19.2 & 4.8 & 3.0 & 7.1 & 3.7 & 3.1 & 2.3 & 1.4 & 1.4 \\
4.8 & 42.4 & 6.4 & 5.4 & 7.5 & 5.0 & 1.8 & 1.7 & 1.7 \\
3.0 & 6.4 & 20.7 & 9.2 & 7.1 & 6.3 & 2.0 & 0.9 & 0.9 \\
7.1 & 5.4 & 9.2 & 16.9 & 10.1 & 6.8 & 3.4 & 1.5 & 1.5 \\
7 & 7.5 & 7.1 & 10.1 & 13.1 & 7.4 & 2.6 & 2.1 & 2.1 \\
3.1 & 5.0 & 6.3 & 6.8 & 7.4 & 10.4 & 3.5 & 1.8 & 1.8 \\
2.3 & 1.8 & 2.0 & 3.4 & 2.6 & 3.5 & 7.5 & 3.2 & 3.2 \\
1.4 & 1.7 & 0.9 & 1.5 & 2.1 & 1.8 & 3.2 & 7.2 & 7.2 \\
1.4 & 1.7 & 0.9 & 1.5 & 2.1 & 1.8 & 3.2 & 7.2 & 7.2
\end{bmatrix}.
$$

In addition, we set $\beta = 0.8$, $\gamma = 0.5$, and $S_i(0), I_i(0), R_i(0) \in [0.1, 10.1]$ ($i = 1, \ldots, 9$).

**Gompertz model** (Gompertz, 1825). The Gompertz model is widely applied in medical research and tumor growth analysis as a kind of growth model. It can be expressed as

$$
\dot{x} = -\theta_1 x \cdot \log(\theta_2 x), \tag{25}
$$

where $\theta_1 = \theta_2 = 1.5$, and $x(0) \in [0.1, 1.1]$.

**Lotka-Volterra equations** (Kingsland, 1995). The Lotka–Volterra equations are used to model the interactions between the predator and prey populations over time, capturing how the population sizes of each species affect the other. This system can be expressed as

$$
\dot{x}_1 = \alpha x_1 - \beta x_1 x_2,
$$
$$
\dot{x}_2 = \delta x_1 x_2 - \gamma x_2, \tag{26}
$$

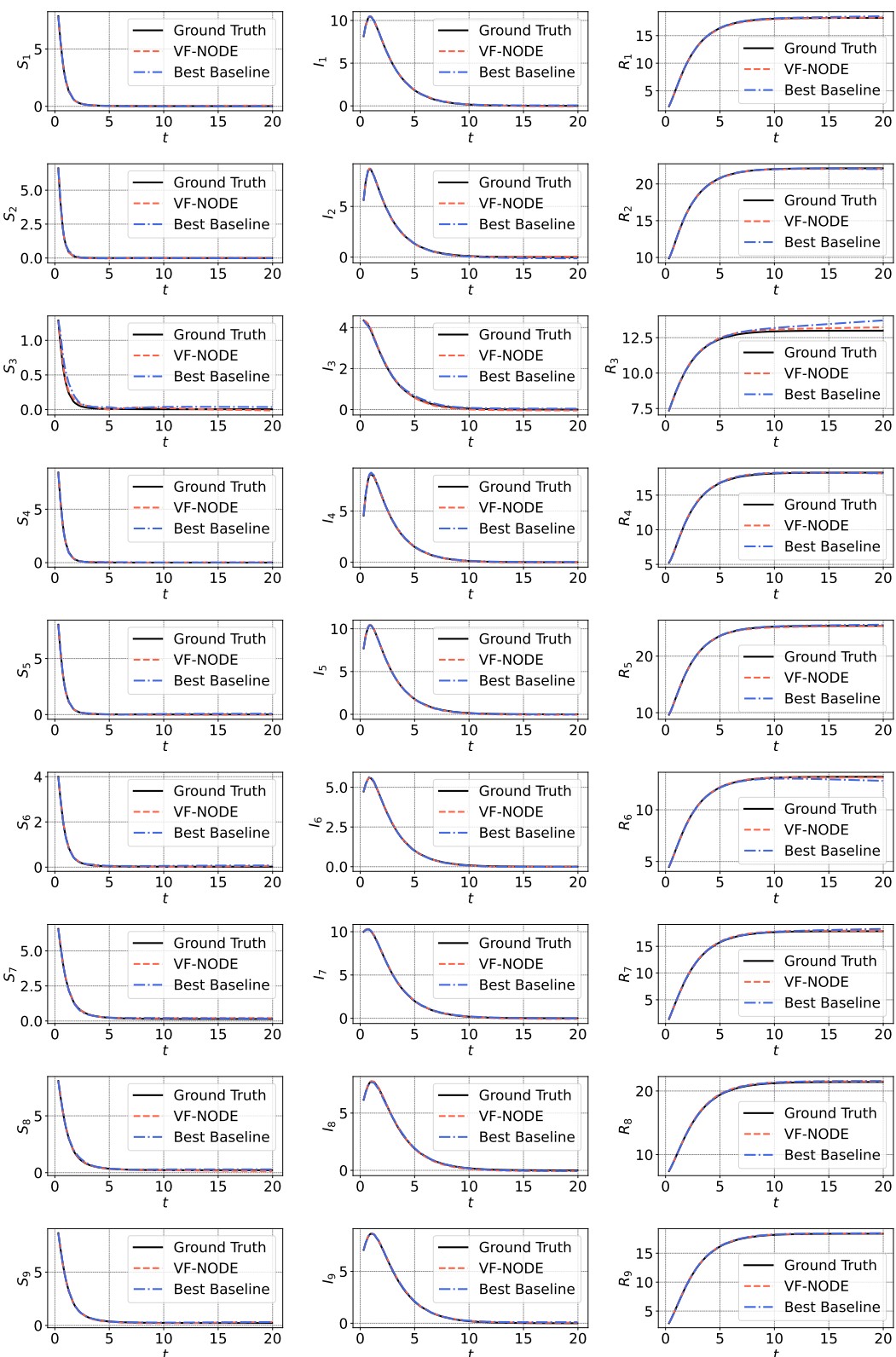

Figure 12: The trajectory plots of the age-structured SIR model with simulation results of VF-NODEs and the best baseline (Vanilla NODE).

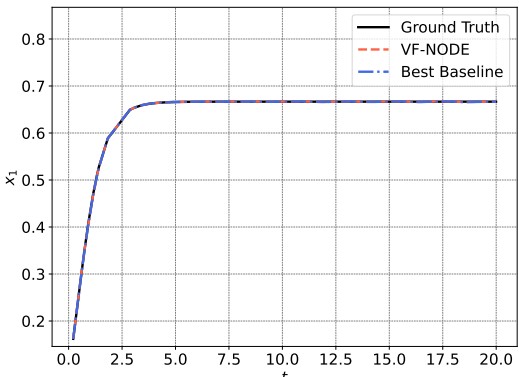

Figure 13: The trajectory plots of the gompertz model with simulation results of VF-NODEs and the best baseline (Vanilla NODE).

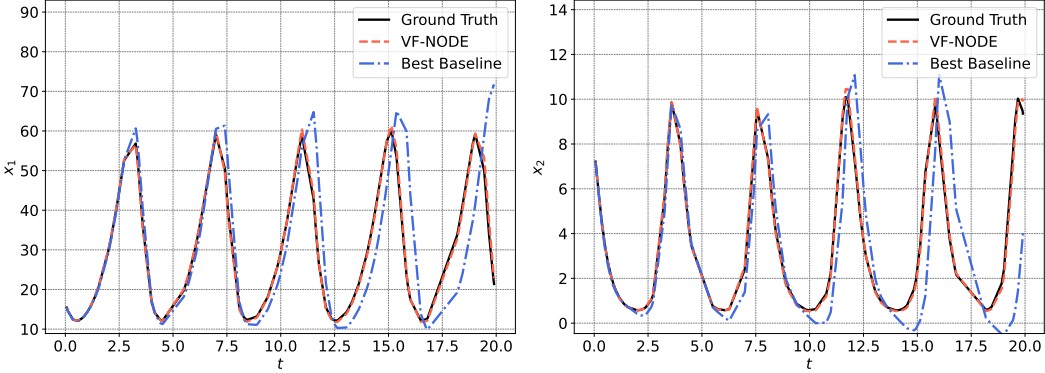

Figure 14: The trajectory plots of the Lotka-Volterra equations with simulation results of VF-NODEs and the best baseline (Vanilla NODE).

where $\alpha = 1.0$, $\beta = 0.3$, $\delta = 0.1$, $\gamma = 3.0$, and $x_1(0) \in [10.0, 20.0]$, $x_2(0) \in [5.0, 10.0]$.

To generate datasets from these dynamical systems, the `Dopri5` solver in `diffrax` is used.

### I.3   TRAINING SETTINGS

For all tasks, we employed the Adam optimizer (Kingma & Ba, 2014), loading all training data in a single epoch. The testing MSE losses of NODE-based baselines reported in all tables were evaluated based on models trained using the optimize-then-discretize approach, following (Kidger, 2022).

For all tasks except modeling the temporal effect of chemotherapy on tumor volume, we set the number of training epochs to 5,000; for the tumor volume modeling task, we used 300 epochs.

For all tasks except the simulation of the age-structured SIR model, we utilized the `cosine_onecycle_schedule` from `optax` as the learning rate scheduler, with an initial learning rate of 0.001. The scheduler parameters were set as follows: `transition_steps` equal to the number of epochs, `peak_value` at 0.01, `pct_start` at 0.2, `div_factor` at 100, and `final_div_factor` at 1,000. For the simulation of the age-structured SIR model, we employed the `cosine_decay_schedule` as the learning rate scheduler, also with an initial learning rate of 0.001. The parameters were set as follows: `decay_steps` equal to the number of epochs, `alpha` at 0.01, and `exponent` at 1.0.

## J   DISCUSSION: APPLYING VF-NODES TO CHAOTIC SYSTEMS

In this section, we discuss the capability of VF-NODEs to model chaotic systems. Since making long-term precise predictions for chaotic systems is nearly impossible, we employ the *unstable periodic orbit* (UPO) detection using *adaptive delayed feedback* (ADF) technique (Zhu et al., 2023) to evaluate whether VF-NODEs can capture the underlying patterns of chaotic systems. Consider the Lorenz system, defined as:

$$
\begin{aligned}
\dot{x} &= \sigma(y - x), \\
\dot{y} &= x(\rho - z) - y, \\
\dot{z} &= xy - \beta z,
\end{aligned}
\tag{27}
$$

where we set $\sigma = 10$, $\beta = 8/3$, and $\rho = 20$, and initial conditions $x(0), y(0), z(0) \in [-10, 10]$. Then, following (Zhu et al., 2023), we apply a control term $c(t) = \gamma(t)e(t)$ to $y$, where $e(t) = y(t - p(t)) - y(t)$. The variables $x$, $y$, $z$, $p$, and $\gamma$ are determined following controlled system:

$$
\begin{aligned}
\dot{x} &= \sigma(y - x), \\
\dot{y} &= x(\rho - z) - y + c(t), \\
\dot{z} &= xy - \beta z, \\
\dot{p} &= r_1[y(t) - y(t - p(t))], \\
\dot{\gamma} &= r_2[y(t) - y(t - p(t))]^2,
\end{aligned}
\tag{28}
$$

where we set $p(0) = \gamma(0) = 0.1$, $r_1 = 0.2$ and $r_2 = 20$. In this setup, $p(t)$ reflects the underlying pattern of the chaotic system. The mean value of $p(t)$ on an interval with the minimum variance can be viewed a UPO for Eq. (27).

These results demonstrate that VF-NODEs can effectively capture the underlying patterns of chaotic systems, significantly outperforming Vanilla NODEs in these challenging scenarios. This improvement is largely due to VF-NODEs mitigating autoregression during the training process, thereby avoiding error accumulation. In contrast, Vanilla NODEs rely heavily on ODE solvers, where error accumulation during training can significantly impact the results.

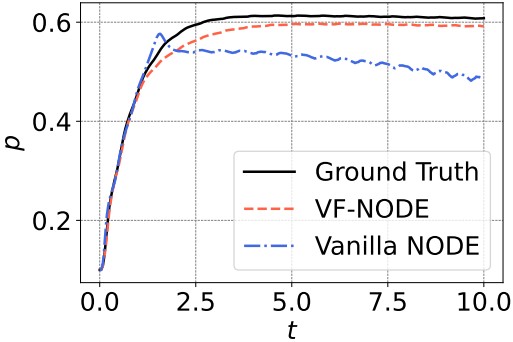

Figure 15: The comparison of $p(t)$ between the VF-NODE and ground truth system in Eq. (28). The detected UPO for the ground truth is $0.6108$, for VF-NODE $(0.5963 \pm 0.0029)$, and for Vanilla NODE $(0.5010 \pm 0.0150)$.

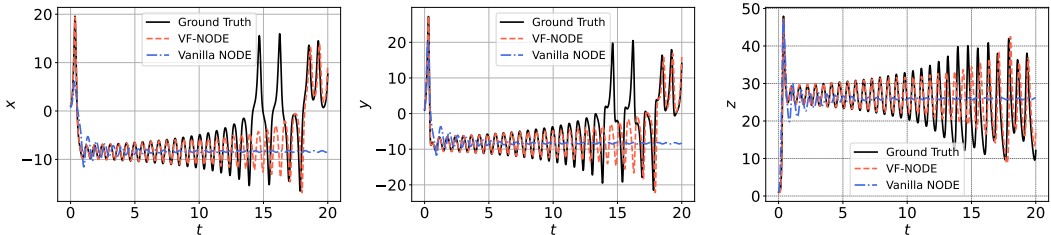

Figure 16: The trajectory plots of the Lorenz system with simulation results of VF-NODEs and Vanilla NODEs. The testing MSE error is $(49.7407 \pm 18.5300)$ for VF-NODEs and for $(54.4111 \pm 2.5761)$ Vanilla NODEs.

