# OpenReview forum: "Accelerating Neural ODEs: A Variational Formulation-based Approach"
_ICLR.cc/2025/Conference — ICLR 2025 Poster_

### Official Review · Reviewer_wbux · 2024-10-18

**Soundness:** 3
**Presentation:** 4
**Contribution:** 3
**Rating:** 6
**Confidence:** 5

**Summary:**

**Edit:** I have increased my score and confidence following the author responses.


This paper proposes a new method for speeding up the training of Neural ODEs. The method removes the need for an ODE solve during the forward and backward pass of a Neural ODE.

The theory behind the approach is to rewrite integration by parts:

$\int_0^T x\dot{\phi} dt = [x\phi]_{0}^{T} - \int_0^T \dot{x}\phi dt$

If $\phi(0)=0$ and $\phi(T)=0$, then $\int_0^T x\dot{\phi} dt + \int_0^T \dot{x}\phi dt = 0$.

The aim is to approximate $\dot{x}$ using a neural network $f_\theta(x, t)$. And so rather than solving the ODE for $f_\theta$ and applying MSE on the observations, the method attempts to minimise $\int_0^T x\dot{\phi_l} dt + \int_0^T f_\theta \phi_l dt$, for a set of orthonormal basis function $\phi_l$ that are zero at the boundaries of the solve.

A fourier basis is used as a natural choice for $\phi_l$, and cubic splines are used to approximate $x(t)$ in the integrals to give them analytical solutions.

Experiments demonstrate good performance measured by MSE, and a speed up in training time.

**Strengths:**

- The writing is strong
- The solution is novel and interesting, it's particularly interesting to see an approach which does not solve the ODE in the forward pass
- The theory behind the method is strong
- The experiments demonstrate good MSE
- The ablations on the basis functions demonstrate using a Fourier basis is a good choice

**Weaknesses:**

- I found the evaluation confusing. As far as I can tell the only place where any wall clock timing is carried out is in Figure 2, on one dataset, the glycolytic model. All other tables seem to be about MSE. However, the main claims and messages of the paper are about accelerating neural ODEs. If the main claim is that this approach speeds up training then this should be shown more across all datasets. Are these numbers available, have I missed them?
- Figure 2 shows no uncertainties on the time taken to train.
- There is an error in line 137-138. If there is regularisation applied during training, the training time can also be reduced, since later iterations are made faster after regularisation is applied in earlier iterations of training.
- Following on from this, the experiments would be made more convincing by testing some of these regularisation methods. For example minimising higher order derivatives. Currently the main baselines for speed are discretise-then-optimise, optimise-then-discretise and Seminorms, of which only Seminorms claim to speed up Neural ODEs. This point would not be as relevant if the main claims of the paper are about MSE rather than time to train.
- It is not clear what ODE solver is used in the experiments, if adaptive solvers are used it is harder to make the argument that the forward solve is slow. The proposed method is linear in the number of observed points (to do cubic interpolation), however if there are many observed points this method will be slower compared to an adaptive solver if the trajectories are quite smooth. Have adaptive solvers been tested? The reverse is also true, if the dynamics are complex an adaptive solver can take smaller steps whereas the proposed method might suffer from accuracy issues as identified in the paper.

**Questions:**

- Is it possible to include two papers in the related work: 1) STEER: Simple Temporal Regularization For Neural
ODEs and 2) Interpolation Technique to Speed Up Gradients Propagation in Neural ODEs?

---

> ### Author Response · Authors · 2024-11-23
> **Rebuttal by Authors**
>
> Dear Reviewer wbux,
>
> We sincerely thank you for taking the time to review our paper and for your thoughtful and insightful comments. We have carefully addressed all the points and suggestions you raised. If you have any additional concerns or questions, please do not hesitate to let us know. We deeply appreciate this opportunity to refine our work and would be truly grateful for any further feedback you could provide. Thank you once again!
>
> **Q 4.1: If the main claim is that this approach speeds up training then this should be shown more across all datasets.**
>
> **A 4.1:**  Thank you for your valuable suggestion! We have included two additional timing test results for the repressilator and the age-structured SIR models in Figs 4, 5 in Appendix F.1 (page 18) to further demonstrate the training speed improvements of our approach. Since both the glycolytic model and the toggle switch model share the same dimensionality of 2, we provided the timing test results only for the glycolytic model to avoid redundancy. Please note that the average training times were evaluated on a different hardware setting compared to the initial submission, which may cause slight variations in the newly reported results.
>
> **Q 4.2: Figure 2 shows no uncertainties on the time taken to train.**
>
> **A 4.2:** We have added uncertainty measurements into all reported training time results, as shown in Figs. 2 (page 8), 4, 5 (page 18).
>
> **Q 4.3: There is an error in line 137-138. If there is regularisation applied during training, the training time can also be reduced, since later iterations are made faster after regularisation is applied in earlier iterations of training.**
>
> **A 4.3:** [1] pointed out the overall training time for regularization methods may not necessarily decrease due to the added computational overhead of the regularizer. This is also verified by our experimental results in Table 4.1 and updated timing test results in Figs 2, 4, 5. Based on your feedback, we have revised our claims to provide a more rigorous statement, as highlighted in blue in lines 138-139 of the updated submission.
>
> [1] Kelly et al. "Learning differential equations that are easy to solve." NeurIPS 2020.
>
> **Q 4.4: The experiments would be made more convincing by testing some of these regularisation methods.**
>
> **A 4.4:** We compared VF-NODEs with TayNODEs [1], which uses high-order derivative regularization. Partial experimental results are shown in the following Table 4.1, with the full results available in all related tables and figures of the updated submission (highlighted in blue). In all experiments, the performance of TayNODEs is significantly lower than that of VF-NODEs. This is due to the high-order derivative penalties, which constrain the complexity of learned dynamics and thereby limit their expressivity of NODEs for time series tasks.
>
> Furthermore, the training speed of TayNODEs is slower than both VF-NODEs and even Vanilla NODEs. This can be attributed to two key factors: (1) TayNODEs still rely on ODE solvers, which generally result in a higher Number of Function Evaluations (NFEs) compared to VF-NODEs, and (2) the additional computational overhead introduced by Taylor-mode automatic differentiation further extends the training time.
>
> **Table 4.1**: Comparison of Testing MSE (mean±standard deviation) and Average Training Time per Epoch (sec) among Vanilla NODEs, TayNODEs and VF-NODEs.
>
> |               | Glycolytic-Interp         | Glycolytic-Extrap         | Toggle-Interp             | Toggle-Extrap             | Average Training Time per Epoch (sec) |
> | :------------ | :------------------------ | :------------------------ | :------------------------ | :------------------------ | ------------------------------------- |
> | Vanilla NODEs | (1.51e-03)±(1.40e-03      | (8.79e-04)±(7.64e-04)     | (8.00e-04)±(8.69e-04)     | (8.14e-07)±(6.72e-07)     | (1.549e-01 ± 1.345e-03)               |
> | TayNODE       | (3.20e-03)±(1.32e-03)     | (4.71e-03)±(3.60e-03)     | (1.37e-02)±(9.96e-03)     | (5.09e-02)±(5.09e-02)     | (4.219e-01 ± 5.073e-03)               |
> | VF-NODE       | **(6.35e-05)±(2.68e-06)** | **(1.63e-04)±(3.05e-05)** | **(1.69e-04)±(6.09e-05)** | **(4.79e-07)±(5.24e-08)** | **(2.270e-02 ± 1.567e-04)**           |
>
> [1] Kelly et al. "Learning differential equations that are easy to solve." NeurIPS 2020.

---

> ### Author Response · Authors · 2024-11-23
> **(Continued) Rebuttal by Authors**
>
> **Q 4.5: It is not clear what ODE solver is used in the experiments, if adaptive solvers are used it is harder to make the argument that the forward solve is slow. The proposed method is linear in the number of observed points (to do cubic interpolation), however if there are many observed points this method will be slower compared to an adaptive solver if the trajectories are quite smooth. Have adaptive solvers been tested? The reverse is also true, if the dynamics are complex an adaptive solver can take smaller steps whereas the proposed method might suffer from accuracy issues as identified in the paper.**
>
> **A 4.5**: As noted in lines 1233-1235, all experimental results use adaptive ODE solvers. The experimental results in Figs. 2, 4, 5 demonstrate our method is still faster than Vanilla NODEs with the discretize-then-optimize approach. Additionally, we make the following arguments:
>
> 1. In the first case, we observe that even with a larger number of sampled points and smooth trajectories, Vanilla NODEs with adaptive solvers can still be slower than our method. To illustrate this, we tested both models on the ODE  $\dot{x}=0.001 x$, which has the analytical and inherently smooth solution $x(t) = x_0 \exp(0.001t)$. For this experiment, we set $T = 100$ with 1000 sampled points in this experiment. with 1000 sampled points. The testing MSE and average training time per epoch are provided in Table 4.2.
>
>    The results highlight that VF-NODEs achieve excellent performance and acceleration, even with the additional computational cost introduced by spline methods. The performance of Vanilla NODEs is still worse than that of VF-NODEs due to the autoregression in ODE solvers. Additionally, beyond the reduced Number of Function Evaluations (NFEs), improved parallelism is also a key factor in this efficiency, as discussed in Section 4.3. In VF-NODEs, the evaluation of DNN-based vector fields is performed simultaneously. In contrast, Vanilla NODEs rely on the autoregressive nature of ODE solvers, requiring sequential evaluations of the DNN. This limitation restricts parallelism and significantly increases computational overhead.
>
> 2. For the second case, we already evaluated VF-NODEs on both a 6-dimensional Repressilator model and a 27-dimensional age-structured SIR model, as shown in Tables 1 and 2 (page 8, 9) of the main paper. Additionally, systems such as the Glycolytic model and the Lotka-Volterra system, which exhibit oscillatory trajectories, were also included in our experiments (trajectory plots are provided in Appendix H.2). These experimental results indicate that VF-NODEs still outperform Vanilla NODEs on complex dynamics.
>
> **Table 4.2**: Comparison of Testing MSE (mean±standard deviation) and Average Training Time per Epoch (sec) of Vanilla NODEs and VF-NODEs on $\dot{x} = 0.001x$.
>
> |              | Interp                | Extrap                | Average Training Time / Epoch |
> | ------------ | --------------------- | --------------------- | ----------------------------- |
> | Vanilla NODE | (5.34e-04)±(3.50e-04) | (9.61e-04)±(5.10e-04) | (4.72e-01)±(2.13e-01)         |
> | VF-NODE      | (5.58e-08)±(7.27e-08) | (5.08e-07)±(6.64e-07) | (4.58e-01)±(2.04e-03)         |
>
> **Q 4.6: Is it possible to include two papers in the related work: 1) STEER, and 2) IRDM?**
>
> **A 4.6:** We now include the suggested papers in the related work, as shown in lines 136-144. STEER [2] is a regularization method that randomly samples the end time of the ODE during training. However, as noted in A 4.4, such regularization approaches still rely on ODE solvers and can restrict the expressivity of NODEs in time series tasks. Additionally, the regularization term may introduce extra computational costs. IRDM [3] leverages barycentric Lagrange interpolation (BLI) to replace the ODE solving process in the adjoint method. However, it still depends on ODE solvers during the forward pass, resulting in a high Number of Function Evaluations (NFEs).
>
> [2] Ghosh et al. "Steer: Simple temporal regularization for neural ode." NeurIPS 2020.
>
> [3] Daulbaev et al. "Interpolation technique to speed up gradients propagation in neural odes." NeurIPS 2020.

---

> ### Author Response · Authors · 2024-11-25
> **Request for Follow-Up Feedback on Author Rebuttal**
>
> Dear Reviewer,
>
> Our detailed rebuttal has been submitted, and we have thoroughly addressed all the points and suggestions you raised. We understand the significant workload involved in reviewing papers, but we kindly request your feedback on our responses to ensure that the discussions are as productive and comprehensive as possible. Your insights will be invaluable in refining the final version of our work. Thank you once again for your time and effort!

---

> ### Comment · Reviewer_wbux · 2024-11-25
> **Thank you for the responses, I have increased my score**
>
> Thank you to the authors for the significant time spent writing responses and making the changes. I have increased my score. Regarding each separate point:
>
> - Time vs MSE: I appreciate including more training times, this makes the results far more in line with the claims and purpose of the paper.
> - Uncertainties: many thanks for including these uncertainties.
> - Regularization's effect on training: This makes sense to me, I suppose it depends on the type of regularization used, "How To Train your Neural ODE" for example uses a term that provides significant overhead, whereas "Opening the Blackbox: Accelerating Neural Differential Equations by Regularizing Internal Solver Heuristics" does not. I appreciate the attention to detail in the updated phrasing.
> - Adding TayNODEs: Many thanks for including a more relevant baseline, the paper would still be improved with more baselines, however I do believe these new results push the paper over the acceptance boundary.
> - Adaptive solvers: My apologies for not spotting the detail on lines 1233-1235, this concern is alleviated.
> - Related work: Many thanks for including these citations.

---

> > ### Author Response · Authors · 2024-11-26
> > **Thanks for raising the score**
> >
> > Dear Reviewer,
> >
> > Thanks for increasing the score. We will integrate all your comments and suggestions into our final version.

---

### Official Review · Reviewer_BidZ · 2024-10-28

**Soundness:** 3
**Presentation:** 3
**Contribution:** 2
**Rating:** 6
**Confidence:** 3

**Summary:**

This paper introduces VF-NODE, a new method for accelerating the training of Neural ODEs by using a variational formulation that evaluates vector fields only at observed data points, reducing function evaluations (NFEs) and eliminating autoregression, thereby minimizing error accumulation. The approach integrates Filon’s method to handle oscillations in the loss function and uses natural cubic spline regression to manage noisy or incomplete data. Experiments show VF-NODE is 10x to 1000x faster than traditional methods while maintaining or improving accuracy.

**Strengths:**

This paper introduces a novel approach using variational formulation (VF) to greatly accelerate Neural ODE training, reducing function evaluations and improving accuracy. By integrating Filon's method with cubic spline regression for handling oscillatory integrals, the method achieves 10x to 1000x faster training across various dynamical systems with higher or competitive accuracy, demonstrating originality, technical rigor, and real-world applicability.

**Weaknesses:**

The author accelerates the training of the original NODEs method by adopting the VF-NODEs approach. However, the necessity and rationality of transforming the training problem into the optimization problem of Equation 6 still require further elaboration.  See Questions for details.

**Questions:**

1. The author utilized natural spline interpolation to fit the orbit **$x$** as well as the vector field $f_\theta$ in this process. Then, why not directly adopt the strategy of gradient flow matching for training (see literature [1]). This approach seems more straightforward and bypasses the high computational costs associated with traditional numerical integration.

2. As the author did not present the trajectory plots in experiments, I am concerned about the performance boundaries of this method in handling time series data. Is this method only effective on data with simple behaviors, or can it still outperform traditional NODE methods for more complex systems (such as chaotic systems) ?

3. When using natural spline interpolation, is there an overfitting issue, such as the Runge phenomenon?

4. This method directly inputs the data from the original system, thus it cannot train through modeling latent variables (like latent ODE method), which may result in a loss of the method's flexibility.

[1] Li X, Zhang J, Zhu Q, et al. From Fourier to Neural ODEs: Flow matching for modeling complex systems[J]. arXiv preprint arXiv:2405.11542, 2024.

If the authors can answer the above questions well, I would be happy to consider raising the score.

---

> ### Author Response · Authors · 2024-11-23
> **Rebuttal by Authors**
>
> Dear Reviewer BidZ,
>
> Thank you for taking the time to review our paper and for your insightful comments. We have carefully addressed all the feedback and suggestions you provided. Please kindly let us know if you have any additional concerns or require further clarifications. We truly appreciate this opportunity to improve our work and are grateful for any further feedback you might offer. Thank you once again!
>
> **Q 3.1 The author utilized natural spline interpolation to fit the orbit $x$ as well as the vector field $f_\theta$ in this process. Then, why not directly adopt the strategy of gradient flow matching for training (see literature [1])?**
>
> **A 3.1:** As noted on Page 4 of [1], a key limitation of gradient matching methods is that numerical derivative estimation is more sensitive to noisy and sparse observations. In fact, [1] highlights that the variational formulation is essentially an improvement and generalization of traditional gradient matching, addressing these sensitivity issues effectively. To demonstrate this, we implemented a gradient matching baseline using natural cubic splines to estimate the derivatives of sampled trajectories at observed points (We did not implement FNODEs suggested by the reviewer, since they are not applicable on irregularly-sampled data). We refer to this method as *Grad-Matching NODEs*. Partial experimental results are shown in Table 3.1, with the full results available in Tables 12 and 13 in Appendix G (page 22) of the updated submission. Although both models have similar training speeds (average training time per epoch around $10^{-2}$ seconds), the accuracy of Grad-Matching NODEs is significantly lower than that of VF-NODEs. This performance gap can be attributed to their sensitivity to noisy and sparse sampling in our experimental settings.
>
> **Table 3.1**: Comparison of Testing MSE (mean±standard deviation) between Grad-Matching NODEs and VF-NODEs.
>
> |                    | Glycolytic-Interp         | Glycolytic-Extrap         | Toggle-Interp             | Toggle-Extrap             |
> | :----------------- | :------------------------ | :------------------------ | :------------------------ | :------------------------ |
> | Grad-Matching NODE | (7.91e-04)±(6.52e-04)     | (6.32e-04)±(1.86e-04)     | (1.19e-03)±(8.55e-04)     | (4.43e-06)±(4.22e-06)     |
> | VF-NODE            | **(6.35e-05)±(2.68e-06)** | **(1.63e-04)±(3.05e-05)** | **(1.69e-04)±(6.09e-05)** | **(4.79e-07)±(5.24e-08)** |
>
> [1] Brunel et al. "Parametric estimation of ordinary differential equations with orthogonality conditions." Journal of the American Statistical Association.
>
> **Q 3.2: As the author did not present the trajectory plots in experiments, I am concerned about the performance boundaries of this method in handling time series data. Is this method only effective on data with simple behaviors, or can it still outperform traditional NODE methods for more complex systems (such as chaotic systems)?**
>
> **A 3.2:** We have now included trajectory plots in Appendix H.2, which illustrate the performance of VF-NODEs. Some of the tested dynamical systems exhibit complex behaviors, such as oscillating trajectories, and VF-NODEs consistently outperform Vanilla NODEs in these scenarios. However, we acknowledge that modeling chaotic systems remains an open challenge due to their inherent sensitivity to initial conditions. While VF-NODEs provide a robust framework, further advancements are needed to fully address the challenges posed by chaotic dynamics. We plan to explore new techniques tailored to these systems in future work.
>
> [2] De Boor, C. "A practical guide to splines." Springer.

---

> ### Author Response · Authors · 2024-11-23
> **(Continued) Rebuttal by Authors**
>
> **Q 3.3: When using natural spline interpolation, is there an overfitting issue, such as the Runge phenomenon?**
>
> **A 3.3:** No. (1) We primarily use spline regression to handle sampled data, which is robust for noisy and sparse data [3]. (2) The Runge phenomenon occurs with high-degree polynomials [4], whereas our spline regression/interpolation uses only 3rd-degree polynomials.
>
> [3] Knowles et al. "Methods for numerical differentiation of noisy data." Electron. J. Differ. Equ 21 (2014): 235-246.
>
> [4] https://en.wikipedia.org/wiki/Runge%27s_phenomenon
>
> **Q 3.4: This method directly inputs the data from the original system, thus it cannot train through modeling latent variables (like latent ODE method), which may result in a loss of the method's flexibility.**
>
> **A 3.4**: Our work mainly focuses on the efficient learning of dynamical systems from irregularly sampled data, especially focusing on scientific fields like chemistry, biology, and epidemiology. This focus aligns with ICLR’s sub-area emphasis on applications in physics and related scientific domains. We acknowledge this as a limitation of our method, as discussed in lines 535-539. However, we believe this limitation could potentially be addressed by incorporating techniques like coordinate gradient descent, as suggested in [5]. We have included a more detailed discussion in line 539 of the updated submission.
>
> [5] Matei et al. "Sensitivity-free gradient descent algorithms." Journal of Machine Learning Research.

---

> ### Author Response · Authors · 2024-11-25
> **Request for Follow-Up Feedback on Author Rebuttal**
>
> Dear Reviewer,
>
> Our detailed rebuttal has been submitted, and we have thoroughly addressed all the points and suggestions you raised. We understand the significant workload involved in reviewing papers, but we kindly request your feedback on our responses to ensure that the discussions are as productive and comprehensive as possible. Your insights will be invaluable in refining the final version of our work. Thank you once again for your time and effort!

---

> ### Comment · Reviewer_BidZ · 2024-11-25
>
> Thank you for the author's response, which has addressed most of my concerns. However, I still believe the behavior demonstrated by the experimental data is overly simplistic, to the extent that it remains unclear whether the method has learned the underlying dynamics or merely performed interpolation fitting. Despite the impossibility of making long-term precise predictions for chaotic systems, their long-term behavioral patterns can be learned through neural network models. For example, reference [1] employs the classical Neural ODE approach to estimate the unstable periodic orbits of chaotic systems. I am curious to understand whether the method proposed by the authors retains its advantages in these more complex systems.
>
> [1] Zhu Q, Li X, Lin W. Leveraging neural differential equations and adaptive delayed feedback to detect unstable periodic orbits based on irregularly sampled time series[J]. Chaos: An Interdisciplinary Journal of Nonlinear Science, 2023, 33(3).

---

> > ### Author Response · Authors · 2024-11-26
> >
> > Dear Reviewer,
> >
> > We have included additional experiments to demonstrate that VF-NODEs can capture the underlying dynamics of chaotic systems and outperform Vanilla NODEs in such tasks. The details of these experiments can be found in Appendix I (page 28).
> >
> > We deeply appreciate the significant workload involved in reviewing papers, and we kindly request your feedback on our responses to ensure that the discussions are as productive and comprehensive as possible. Your insights are invaluable in helping us refine the final version of our work. Thank you once again for your time and effort!
> >
> > [1] Zhu Q, Li X, Lin W. Leveraging neural differential equations and adaptive delayed feedback to detect unstable periodic orbits based on irregularly sampled time series[J]. Chaos: An Interdisciplinary Journal of Nonlinear Science, 2023, 33(3).

---

> ### Author Response · Authors · 2024-11-26
> **Testing on Chaotic Systems**
>
> **Q 3.5: Despite the impossibility of making long-term precise predictions for chaotic systems, their long-term behavioral patterns can be learned through neural network models. For example, reference [1] employs the classical Neural ODE approach to estimate the unstable periodic orbits of chaotic systems. I am curious to understand whether the method proposed by the authors retains its advantages in these more complex systems.**
>
> **A 3.5**: Thank you for your valuable feedback. Based on the method in [1], we have tested VF-NODEs on the Lorenz system, as shown in Appendix I (page 28). The result has demonstrated our method can capture underlying patterns of chaotic systems. If you have any other concerns, please feel free to tell us. Thank you once again!
>
> [1] Zhu Q, Li X, Lin W. Leveraging neural differential equations and adaptive delayed feedback to detect unstable periodic orbits based on irregularly sampled time series[J]. Chaos: An Interdisciplinary Journal of Nonlinear Science, 2023, 33(3).

---

> ### Comment · Reviewer_BidZ · 2024-11-27
>
> I appreciate the efforts made by the authors during the rebuttal process. Upon reviewing the supplemental experiments on the Lorenz system provided by the authors, it remains unclear how the proposed method performs in predicting more complex behaviors. If the authors could include code in supplementary material demonstrating the proposed method's predictive experiment on the Lorenz system, along with the ground truth and predicted trajectories, I would be inclined to increase my score. However, due to the absence of **credible experimental results** under more complex behaviors, I am unable to assign a positive score at this time.

---

> > ### Author Response · Authors · 2024-11-28
> >
> > Dear Reviewer,
> >
> > We also include the trained models in the supplementary materials. You can load these checkpoints in adf.ipynb in the Anonymous GitHub link to verify the experimental results about Lorenz system.

---

> > > ### Comment · Reviewer_BidZ · 2024-11-29
> > >
> > > I appreciate the authors' efforts regarding this question. After reviewing the updated manuscript, I am willing to increase my score from 5 to 6.

---

> > > > ### Author Response · Authors · 2024-11-29
> > > >
> > > > Dear Reviewer,
> > > >
> > > > Thanks for increasing the score. We will integrate all your comments and suggestions into our final version.

---

> ### Author Response · Authors · 2024-11-28
>
> Dear Reviewer,
>
> We have included trajectory prediction results in Appendix I (page 29). The proposed VF-NODEs still outperform Vanilla NODEs since VF-NODEs mitigate the error accumulation. We also release our code for the Lorenz system in the Anonymous GitHub:
>
> https://anonymous.4open.science/r/VF-NODE-Rebuttal-837B/README.md
>
> Following the README, you can get all the experimental results about chaotic systems in our paper.

---

### Official Review · Reviewer_Ycyv · 2024-11-02

**Soundness:** 3
**Presentation:** 3
**Contribution:** 2
**Rating:** 8
**Confidence:** 3

**Summary:**

This paper proposes a new way to train NeuralODEs that is faster than current approaches. Current training strategies require numerically solving the differential equation which is very computationally expensive due to the high number of function evaluations at each time step. The approach introduced here relies on the variational formulation as a surrogate objective. However, the variational formulation still requires the value of the vector field and underlying time series at each time step, so the authors approximate this with cubic splines. To compute oscillatory integrals, the authors incorporate the Filon method. Finally, the authors demonstrate the performance of their method and show impressive computational time gains. Some prediction performance gains are also observed, due to the non-accumulation of errors in their training approach.

**Strengths:**

Speeding up the training of neural ODE methods is a very important challenge with potentially high impact.

The method proposed shows clear computational time improvements as well as some prediction performance gains.

The method makes sense, and the paper is well written and easy to read.

**Weaknesses:**

One significant weakness that I see with this approach is that it requires defining the trajectories of the time series a priori (here with cubic splines). As such, the model with learn a vector field that agrees with the cubic spline interpolation and will not figure out alternative dynamics. If this is correct, this is a signicant limitation as one important application of NeuralODEs is for data imputation. I would like the authors to discuss that limitation more clearly in the paper and/or to argue against the reasoning above.

Given that you use cubic splines to interpolate the dynamics (and vector field), another baseline can now be considered. That is, you can now use the same cubic spline acceleration to directly integrate Equation 1 and compute the MSE at the observation points. Is such an approach reasonable ? This should be considered as an additional baseline, that doesn't use the variational approach but the cubic spline interpolation.

**Questions:**

As stated above, I would like the authors to discuss the limitations of using an explicit interpolation of the dynamics before training.

4.2 Step 4, the fact of interpolating the vector field using the values only at the observation points seems strange to me as the interpolation will not coincide between observations. That is, between observations, with $\hat{f}$ the interpolation of the vector field:
$ \hat{f}(t) \neq f(\hat{x}(t))$. Can the authors give more details about this discrepancy and motivate why it makes sense ?

During training, your method requires computing cubic splines coefficients for each time series every time (at least for the $b$ coefficients). Can the author elaborate on the compuational cost of such a procedure ? I would also like the authors to explain how the gradient with respect to $\theta$ can still flow from the computation of the $b$ coefficients - how is this end-to-end differentiable ?

---

> ### Author Response · Authors · 2024-11-23
> **Rebuttal by Authors**
>
> Dear Reviewer Ycyv,
>
> Thank you for taking the time to review our paper and for your insightful comments. We have addressed all the suggestions you provided and believe these changes have strengthened our work. Please let us know if you have any additional concerns. We truly appreciate this opportunity to improve our paper and would be grateful for any further feedback you might have. Thank you once again!
>
> **Q 2.1: It requires defining the trajectories of the time series a priori (here with cubic splines). As such, the model with learn a vector field that agrees with the cubic spline interpolation and will not figure out alternative dynamics. If this is correct, this is a signicant limitation as one important application of NeuralODEs is for data imputation.**
>
> **A 2.1:**
>
> 1. We would like to clarify that VF-NODEs do not define trajectories or vector fields *a priori* as cubic splines. Instead, spline methods serve as intermediate steps for Filon's method to compute integrals in the VF loss (Eq. (8)). This approach is *very common* in numerical integration techniques [1], where, based on Newton-Cotes formulas [2], polynomials (such as splines) provide smooth mathematical expressions to *approximate the integrand* from discrete sampled points. The integral is then estimated based on the integral of these approximate polynomials.
> 2. Natural cubic splines are broadly suitable for approximating a broad class of functions, providing accurate approximation for integral computation [1]. Based on our understanding, splines may struggle to capture dynamics when the sampling is extremely sparse. However, this limitation is inherent to all methods including baselines and stems from the Nyquist sampling theorem [3]. In such cases, critical information about the underlying dynamics may be lost, meaning that even Vanilla NODEs would face similar limitations in learning accurate dynamics.
> 3. As noted in lines 407-409, the inference phase of VF-NODEs still relies on ODE solvers. Once trained, VF-NODEs can effectively perform data imputation by leveraging the learned vector fields to infer missing data points.
>
> We think this concern may have been caused by a typo in Fig. 1, where we labeled the "Closed-form Approximation of Estimated Trajectory" as the "Closed-form Approximation of DNN-based Vector Fields" by accident. This typo has now been corrected in the updated Fig. 1.
>
> [1] Carnahan et al. Applied numerical methods. Vol. 2. New York: Wiley, 1969, Chap 2.
>
> [2] https://en.wikipedia.org/wiki/Newton%E2%80%93Cotes_formulas
>
> [3] https://en.wikipedia.org/wiki/Nyquist%E2%80%93Shannon_sampling_theorem
>
> **Q 2.2: New baseline: use the same cubic spline acceleration to directly integrate Equation 1 and compute the MSE at the observation points.**
>
> **A 2.2:** Per your suggestion, we implemented this baseline, named *Spline-Integ NODE*, in the experiment. A portion of the experimental results is shown in Table 2.1, with the full results available in Tables 12 and 13 in Appendix G (page 22) of the updated submission. The training speed of both models is similar (average training time per epoch of approximately $10^{-2}$ seconds), but the accuracy of Spline-Integ NODEs is significantly lower than that of VF-NODEs. This discrepancy arises because the training of Spline-Integ NODEs essentially involves an autoregressive process:
> $$
> \hat{x}(t_{k+1})=\hat{x}(t_k)+\int_{t_k}^{t_{k+1}}f_\theta(\tau,\hat{x}(\tau))d\tau\approx\hat{x}(t_k)+\int_{t_k}^{t_{k+1}}[b_{k, 0}+\sum_{m=1}^3b_{k,m}(\tau-t_k)^m]d\tau=\hat{x}(t_k)+\sum_{m=0}^3b_{k, m}\frac{(t_{k+1}-t_k)^{m+1}}{m+1}
> $$
> where errors accumulate across iterations as each step depends on the approximation from the previous one. In contrast, VF-NODEs rely solely on global numerical integrals, which avoid error accumulation.
>
> **Table 2.1**: Comparison of Testing MSE (mean±standard deviation) between VF-NODEs and Spline-Integ NODEs.
>
> |                   | Glycolytic-Interp         | Glycolytic-Extrap         | Toggle-Interp             | Toggle-Extrap             |
> | :---------------- | :------------------------ | :------------------------ | :------------------------ | :------------------------ |
> | Spline-Integ NODE | (1.34e-03)±(1.44e-03)     | (5.44e-03)±(6.45e-03)     | (9.58e-03)±(7.76e-03)     | (5.22e-05)±(3.55e-05)     |
> | VF-NODEs          | **(6.35e-05)±(2.68e-06)** | **(1.63e-04)±(3.05e-05)** | **(1.69e-04)±(6.09e-05)** | **(4.79e-07)±(5.24e-08)** |

---

> > ### Author Response · Authors · 2024-11-30
> >
> > Dear Reviewer,
> >
> > We are still looking forward to your valuable feedback. If you have any other concerns, please do not hesitate to let us know. You feedback will immensely improve the quality of our work.

---

> ### Author Response · Authors · 2024-11-23
> **(Continued) Rebuttal by Authors**
>
> **Q 2.3: The limitations of using an explicit interpolation of the dynamics before training.**
>
> **A 2.3:** First, we reiterate that spline methods are used to provide accurate approximations with mathematical expressions for numerical integration [1], and they work well in most cases [4], as demonstrated by our experimental results. However, we already explicitly discussed the limitations of spline methods, as noted in lines 535-539, particularly for approximating complex trajectories in intricate dynamical systems (e.g., chaotic systems) or under extremely sparse sampling conditions. However, we argue that this challenge is not unique to our approach; it remains an open issue in the broader community. We will try to explore new techniques to address the limitation in future work.
>
> [1] Carnahan et al. Applied numerical methods. Vol. 2. New York: Wiley, 1969, Chap 2.
>
> [4] De Boor, C. "A practical guide to splines." Springer.
>
> **Q 2.4: Sec 4.2 Step 4, the fact of interpolating the vector field using the values only at the observation points seems strange to me as the interpolation will not coincide between observations. That is, between observations, with $\hat{f}$ the interpolation of the vector field: $\hat{f}(t) \ne f(\hat{x}(t))$. Can the authors give more details about this discrepancy and motivate why it makes sense?**
>
> **A 2.4:**
>
> 1. We emphasize that interpolation techniques are inherently used on discrete observed points [Chap 3.3 of 1]. In VF-NODEs, we only have values of the DNN-based vector fields at observed points, as the trajectories at other time points are unknown. Therefore, using interpolation solely at observed points is a natural and practical choice.
> 2. The discrepancy between the ground truth and the interpolated results is very tiny in most cases [4]. Spline methods typically approximate the original function very accurately. While they may struggle with extremely sparse sampling, this is an open challenge for all other methods including baselines, not just our approach.
> 3. Even though there may be some tiny discrepancy, spline methods still make sense in this context. As noted in A 2.1, they serve as *intermediate steps* of Filon's method for computing integrals numerically  in the VF loss (Eq. (8)), and this is very common in a range of numerical integration techniques [1]. Such polynomial-based numerical integration methods have been well-established in various engineering fields, and our experimental results confirm their effectiveness in VF-NODEs.
>
> [1] Carnahan et al. Applied numerical methods. Vol. 2. New York: Wiley, 1969, Chap 2.
>
> [4] De Boor, C. "A practical guide to splines." Springer.
>
> **Q 2.5: During training, your method requires computing cubic splines coefficients for each time series every time (at least for the $b$ coefficients). Can the author elaborate on the compuational cost of such a procedure?**
>
> **A 2.5:** The computational cost of calculating the spline coefficients primarily arises from solving the associated tridiagonal linear equations [4, 5]. The computational complexity of this process is $\mathcal{O}(N)$.
>
> [4] De Boor, C. "A practical guide to splines." Springer.
>
> [5] https://github.com/jax-ml/jax/discussions/10339
>
> **Q 2.6: I would also like the authors to explain how the gradient with respect to $\theta$ can still flow from the computation of the coefficients - how is this end-to-end differentiable?**
>
> **A 2.6:** The main computational map can be concluded as the following three steps: (1) the DNN vector fields are evaluated on estimated trajectory $\hat{x}(t_k)$, yielding $f_\theta(t_k, \hat{x}(t_k))$. (2) We solve a tridiagonal linear system to get the spline coefficients $b_{k, m}$, and this process is also differentiable [5]. (3) We then compute the VF loss in Eq. (8) based on $a_{k,m}$ and $b_{k,m}$ as shown in Eq. (9), where the integrals can be computed directly without numerical methods.
>
> [5] https://github.com/jax-ml/jax/discussions/10339

---

> > ### Comment · Reviewer_Ycyv · 2024-12-03
> > **Thank you for your response**
> >
> > I want to thank the authors for their careful and thorough response to my comments.
> >
> > In particular, I appreciate the inclusion of the suggested baseline which further shows the benefits of this approach.
> >
> > Some remarks though:
> >
> > `Based on our understanding, splines may struggle to capture dynamics when the sampling is extremely sparse. However, this limitation is inherent to all methods including baselines and stems from the Nyquist sampling theorem` : A common use case for Neural ODE methods would be to learn meaningful interpolations in these extremely sparse scenarios, by relying on amortization over many such sparse time series in the dataset. Using the examples in the dataset, the model would have learn to fill the gaps even in the sparse scenarios. This is the use case I had in mind in my review.
> >
> > That said, I still think the method is interesting, and this particular solution should be of interest to a the community.
> >
> > I raised my score accordingly.

---

> > > ### Author Response · Authors · 2024-12-03
> > > **Thank you for raising the score**
> > >
> > > Dear Reviewer,
> > >
> > > We would like to thank you for your insightful comments and suggestions for improving our paper. We will integrate your comments and suggestions into our revised version. Again, thank you very much!

---

> ### Author Response · Authors · 2024-11-25
> **Request for Follow-Up Feedback on Author Rebuttal**
>
> Dear Reviewer,
>
> Our detailed rebuttal has been submitted, and we have thoroughly addressed all the points and suggestions you raised. We understand the significant workload involved in reviewing papers, but we kindly request your feedback on our responses to ensure that the discussions are as productive and comprehensive as possible. Your insights will be invaluable in refining the final version of our work. Thank you once again for your time and effort!

---

> ### Author Response · Authors · 2024-11-29
>
> Dear Reviewer,
>
> We have received feedback from the other reviewers and kindly request your feedback as well. Your feedback would be immensely helpful in further refining our work. If you have any additional concerns or questions, please do not hesitate to let us know. Your insights and perspective are highly valued, and we sincerely thank you for your time and effort.

---

> ### Author Response · Authors · 2024-11-29
>
> Dear Reviewer,
>
> We are still looking forward to your valuable feedback. If you have any other concerns, please do not hesitate to let us know. You feedback will immensely improve the quality of our work.

---

> ### Author Response · Authors · 2024-12-02
>
> Dear Reviewer,
>
> We are still looking forward to your valuable feedback. If you have any other concerns, please do not hesitate to let us know. You feedback will immensely improve the quality of our work.

---

### Official Review · Reviewer_1H9x · 2024-11-03

**Soundness:** 3
**Presentation:** 3
**Contribution:** 2
**Rating:** 6
**Confidence:** 4

**Summary:**

The authors present a new method for training Neural ODEs (NODEs) based on a variational formulation of an ODE loss. The resulting method used spline regression and to interpolate noisy data which allows for a computation of a variational loss. The advantage of the method is many few function evaluations during training.

**Strengths:**

The strengths of the paper are:
- the proposed method introduces a reasonable approach to training a NODE based on a variational formulation of the loss
- The training method requires many fewer function evaluations, allowing for much faster training than existing methods.
- The method maintains or at times outperms competing methods in terms of accuracy.

**Weaknesses:**

The weaknesses of the paper are:
- Many of the examples seem somewhat toy. It is unclear how the method might extend to much noisier / less structured ODEs with more complex dynamics. In particular one would expect the  spline regression / interpolation to eventually fail on very long, non-smooth or noisy trajectories.
From the paper it is difficult to get a sense of how much these attributes are present in the given benchmarks.
- Implementation of the method is not straightforward from a practitioner's point of view.
- The method introduces a number of hyper parameters required for accurate interpolation which must be chosen/tuned.
- The benefit of reduced training, while convenient, is not broadly important for many of the problems considered.

**Questions:**

In the COVID-19 dataset example is the data gathered from real world observations? Or is the data generated from some parametric model of COVID-19 spread?

Do the authors have a sense of the limits of the spline regression / interpolation? On what sorts of trajectories it might fail?

It would be very helpful if the authors could provide plots of training trajectories the ODEs in consideration so the reader could assess the noise levels / complexity of the trajectories.

---

> ### Author Response · Authors · 2024-11-23
> **Rebuttal by Authors**
>
> Dear Reviewer 1H9x,
>
> We sincerely thank you for taking the time to review our paper and for providing insightful comments. We have carefully addressed all the points and suggestions you raised. Please feel free to let us know if you have any additional concerns or questions. We genuinely appreciate this opportunity to refine our work and would be grateful for any further feedback you could provide. Thank you once again!
>
> **Q 1.1: It is unclear how the method might extend to much noisier / less structured ODEs. Spline methods may fail on very long, non-smooth or noisy trajectories.**
>
> **A 1.1:**
>
> 1. We already evaluated VF-NODEs on both a 6-dimensional repressilator model and a 27-dimensional age-structured SIR model, as shown in Tables 1 and 2 (page 8, 9) of the updated submission. Additionally, the detailed trajectories of these systems, included in Appendix H.2, also demonstrate intricate behaviors, including oscillatory trajectories. These results illustrate VF-NODE’s scalability to higher dimensions and ability to handle intricate dynamics.
> 2. We have conducted experiments to examine the impact of noise and sampling on our method, as detailed in Appendix F.4 (page 19) of the submission. To further address your concerns, we conducted additional experiments with increased relative noise levels, $\sigma_R = 0.1, 0.3, 0.5$ and extended time durations, $T = 10, 20, 50$. The results, as shown in Tables 1.1 and 1.2, demonstrate that VF-NODEs can maintain strong performance across diverse settings.
>
> **Table 1.1**: Testing MSE (mean±standard deviation) on the effect of relative noise level.
>
> | $\sigma_R$ | Glycolytic-Interp     |   Glycolytic-Extrap   |     Toggle-Interp     |     Toggle-Extrap     |
> | :--------: | --------------------- | :-------------------: | :-------------------: | :-------------------: |
> |    0.1     | (1.08e-03)±(2.89e-04) | (1.57e-03)±(1.46e-04) | (2.10e-02)±(2.78e-02) | (3.48e-02)±(4.85e-02) |
> |    0.3     | (1.69e-02)±(9.83e-03) | (3.86e-02)±(1.70e-02) | (4.68e-02)±(2.45e-03) | (1.89e-01)±(3.33e-02) |
> |    0.5     | (5.78e-02)±(3.09e-02) | (9.58e-02)±(4.13e-02) | (1.34e-01)±(8.08e-02) | (7.29e-01)±(1.13e-01) |
>
> **Table 1.2**: Testing MSE (mean±standard deviation) on the effect of time duration $T$.
>
> | Time Duration $T$ |   Glycolytic-Interp   |   Glycolytic-Extrap   |     Toggle-Interp     |     Toggle-Extrap     |
> | :---------------: | :-------------------: | :-------------------: | :-------------------: | :-------------------: |
> |        10         | (6.35e-05)±(2.68e-06) | (1.63e-04)±(3.05e-05) | (1.69e-04)±(6.09e-05) | (4.79e-07)±(5.24e-08) |
> |        20         | (4.11e-04)±(4.60e-04) | (9.27e-04)±(8.23e-04) | (1.24e-04)±(6.65e-05) | (2.28e-07)±(4.25e-08) |
> |        50         | (3.22e-04)±(1.21e-05) | (2.34e-03)±(9.19e-04) | (3.86e-04)±(3.18e-04) | (1.90e-07)±(8.22e-08) |
>
> **Q 1.2: Implementation of the method is not straightforward from a practitioner's point of view.**
>
> **A 1.2:** The key steps of VF-NODE are outlined in Figure 1 (page 2), Section 4.2 (page 6), and the detailed process is presented in Algorithm 1 (page 18). From an implementation perspective, VF-NODEs rely only on the VF loss, which is composed of a series of integrals. To compute these integrals, spline methods are used to provide accurate approximations for Filon’s method. These components are straightforward to implement. We will also release our code publicly upon publication.

---

> ### Author Response · Authors · 2024-11-23
> **(Continued) Rebuttal by Authors**
>
> **Q 1.3: The method introduces a number of hyper parameters required for accurate interpolation which must be chosen/tuned.**
>
> **A 1.3**: In fact, according to Algorithm 1 (page 18) in the updated submission, the proposed VF-NODEs only introduce *two* new hyper-parameters: (1) the number of basis functions $L$, and (2) the hyper-parameter $\lambda$ in spline regression. Both hyperparameters are straightforward to choose and tune, as the performance of VF-NODEs is not sensitive to them:
>
> 1. **Number of Basis Functions $L$**. Fig. 6 in Appendix F.4 (page 20) of submission has demonstrated that VF-NODE performance remains stable across various values. We also conducted additional experiments to test the influence of $L$ on the VF-NODEs' training speed, updated in Fig. 7 in Appendix F.4 (page 20). These results confirm that $L$ has minimal impact on both the performance and training speed of VF-NODEs.
>
> 2. **Spline Regression Parameter $\lambda$**. Typically, $\lambda$ values greater than 0.9 are suitable for most cases. Table 1.3 presents an ablation study on the sensitivity of VF-NODEs to $\lambda$, which demonstrate that our method is not sensitive to $\lambda$.
>
> **Table 1.3:** Ablation Study on the Sensitivity of VF-NODEs to  $\lambda$.
>
> |  $\lambda$ | Glycolytic-Interp     | Glycolytic-Extrap     | Toggle-Interp         | Toggle-Extrap         |
> | :------------- | :-------------------- | :-------------------- | :-------------------- | :-------------------- |
> |0.9     | (4.20e-03)±(3.44e-04) | (4.81e-03)±(6.03e-04) | (5.40e-03)±(2.46e-04) | (7.57e-06)±(5.49e-06)|
> |0.99    | (1.14e-03)±(2.00e-04) | (1.78e-03)±(7.26e-04) | (2.16e-03)±(7.42e-04) | (8.29e-06)±(9.32e-06)|
> |0.999   | (7.27e-04)±(6.59e-04) | (7.80e-04)±(5.77e-04) | (1.79e-03)±(9.64e-04) | (1.52e-05)±(9.32e-06)|
> |0.9999  |(5.32e-04)±(4.47e-04) | (1.41e-03)±(1.59e-03) | (1.24e-03)±(2.33e-04)  |(7.87e-06)±(3.23e-06) |
> |0.99999 | (6.35e-05)±(2.68e-06) | (1.63e-04)±(3.05e-05) | (1.69e-04)±(6.09e-05) | (4.79e-07)±(5.24e-08)|
>
> **Q 1.4: The benefit of reduced training, while convenient, is not broadly important for many of the problems considered.**
> **A 1.4:** Our work not only reduces the training time of NODEs but also enhances accuracy in learning dynamical systems from irregularly sampled data, making our method more practical and impactful in resource-constrained settings like IoT edge computing compared to Vanilla NODEs. Moreover, our approach is better suited for complex tasks in fields like chemistry, biology, and epidemiology [1], where the slow training process and error accumulation of vanilla NODEs have limited their performance.
>
> [1] Ram et al. "A modified age-structured SIR model for COVID-19 type viruses." Scientific reports 2021.
>
> **Q 1.5: In the COVID-19 dataset example is the data gathered from real world observations? Or is the data generated from some parametric model of COVID-19 spread?**
>
> **A 1.5:** As noted in line 431, the COVID-19 dataset is based on real-world observations and sourced from the COVID-19 Data Hub [2].
>
> [2] Guidotti et al. "COVID-19 data hub." Journal of Open Source Software.
>
> **Q 1.6: Do the authors have a sense of the limits of the spline regression/interpolation? On what sorts of trajectories it might fail?**
>
> **A 1.6:** First, we emphasize that spline methods are used mainly as intermediate steps to compute integrals in the VF loss. These methods generally provide accurate approximations with mathematical expressions [3, 4] for numerical integration, as demonstrated by our experimental results. However, as noted in lines 535-539, we acknowledge that splines may encounter limitations when modeling particularly complex trajectories—especially in chaotic systems or under extremely sparse sampling conditions. We argue that these challenges are fundamental open problems in dynamical systems modeling and are not unique to our approach. We plan to investigate alternative techniques in future work.
>
> [3] De Boor, C. "A practical guide to splines." Springer.
>
> [4] Carnahan et al. Applied numerical methods. Vol. 2. New York: Wiley, 1969, Chap 2.
>
> **Q 1.7: It would be very helpful if the authors could provide plots of training trajectories the ODEs.**
>
> **A 1.7:** Per your suggestion, we have provided plots of both the ground truth trajectories and simulated trajectories of VF-NODEs and the best baselines for each system we tested in Appendix  H.2.

---

> ### Author Response · Authors · 2024-11-25
> **Request for Follow-Up Feedback on Author Rebuttal**
>
> Dear Reviewer,
>
> Our detailed rebuttal has been submitted, and we have thoroughly addressed all the points and suggestions you raised. We understand the significant workload involved in reviewing papers, but we kindly request your feedback on our responses to ensure that the discussions are as productive and comprehensive as possible. Your insights will be invaluable in refining the final version of our work. Thank you once again for your time and effort!

---

> ### Author Response · Authors · 2024-11-26
> **Test on Chaotic Systems**
>
> Dear Reviewer,
>
> To further address your concern in Q 1.1, we have conducted additional experiments with VF-NODEs on the chaotic Lorenz system, incorporating suggestions from Reviewer BidZ as discussed in Q 3.5. The results are presented in Appendix I (page 28) and demonstrate that VF-NODEs effectively capture the underlying patterns of chaotic systems.
>
> We understand the significant workload involved in reviewing papers, but we kindly request your feedback on our responses to ensure that the discussions are as productive and comprehensive as possible. Your insights are invaluable in refining the final version of our work. Thank you once again for your time and effort!

---

> > ### Comment · Reviewer_1H9x · 2024-11-26
> >
> > I appreciate the authors thorough response. Some of my concerns have been addressed.
> >
> > After reviewing the trajectory plots, I agree with some of my fellow reviewers [see BidZ] that some of the examples are still quite simplistic. For example in the SIR model many of the dimensional look almost identical. Additionally many of the other examples demonstrate relatively simple oscillatory behavior.
> >
> > I still feel the authors have improved the paper and thus I will raise my score.

---

> > > ### Author Response · Authors · 2024-11-26
> > > **Thanks for raising the score**
> > >
> > > Dear Reviewer,
> > >
> > > Thanks for raising the score. We will integrate your comments and suggestions into our revised version.

---

### Author Response · Authors · 2024-11-23
**General Response to All Reviewers**

Dear Reviewers,

We sincerely thank you for taking the time to review our paper and for providing insightful comments. We have carefully addressed all the points and suggestions you raised and successfully **submitted a revised manuscript**. Notably, we have made significant improvements in the following areas:
1. Inclusion of TayNODEs [1] as a new baseline in all experiments.
2. Additional timing test results on the repressilator model and the age-structured SIR model (Appendix F.1).
3. Ablation studies on the sensitivity of VF-NODEs to other hyperparameters and the effect of the sampling interval (Appendix F.4).
4. Discussion of alternative training methods for NODEs based on spline methods (Appendix G).
5. Trajectory plots have been included in Appendix H.2.

We have **highlighted these significant modifications in blue** throughout the manuscript. Please do not hesitate to let us know if you have any additional concerns or questions. We genuinely appreciate this opportunity to refine our work further and would be grateful for any additional feedback you could provide. Thank you once again!

[1] Kelly et al. "Learning differential equations that are easy to solve." NeurIPS 2020.

---

### Meta-Review · Area_Chair_MS8D · 2024-12-22

**Metareview:**

The paper focuses on Neural ODEs, the topic that is slightly out of focus nowdays  for ML conferences due to the rise of diffusion models and flow matching, but still potentially useful subject of study. Specifically, the NODEs right-hand side is trained to fit an irregular time series by using a variational formulation of the ODE, which allows for much faster training. All reviewers are supportive for the work, and I agree with them. A minor comment that is missing from the text is that NODE are used not only to approximate dynamical systems, but are also used for generic tasks such as classification or generative modelling. It is unclear, wether the proposed approach can be used for those tasks.

**Additional Comments On Reviewer Discussion:**

The discussion mainly consisted in adding new, challenging experiments and the authors indeed have increased the amount of examples in the paper.

---

### Decision · Program_Chairs · 2025-01-22

Accept (Poster)